# RNF43 truncations trap CK1 to drive niche-independent self-renewal in cancer

Maureen Spit[1,†] (ID), Nicola Fenderico[1,†] (ID), Ingrid Jordens[1,†] (ID), Tomasz Radaszkiewicz[2] (ID), Rik GH Lindeboom[3] (ID), Jeroen M Bugter[1] (ID), Alba Cristobal[1], Lars Ootes[1] (ID), Max van Osch[1], Eline Janssen[1], Kim E Boonekamp[1], Katerina Hanakova[4], David Potesil[4], Zbynek Zdrahal[4], Sylvia F Boj[5], Jan Paul Medema[6] (ID), Vitezslav Bryja[2] (ID), Bon-Kyoung Koo[7] (ID), Michiel Vermeulen[3] (ID) & Madelon M Maurice[1,*] (ID)

## Abstract

Wnt/β-catenin signaling is a primary pathway for stem cell maintenance during tissue renewal and a frequent target for mutations in cancer. Impaired Wnt receptor endocytosis due to loss of the ubiquitin ligase RNF43 gives rise to Wnt-hypersensitive tumors that are susceptible to anti-Wnt-based therapy. Contrary to this paradigm, we identify a class of RNF43 truncating cancer mutations that induce β-catenin-mediated transcription, despite exhibiting retained Wnt receptor downregulation. These mutations interfere with a ubiquitin-independent suppressor role of the RNF43 cytosolic tail that involves Casein kinase 1 (CK1) binding and phosphorylation. Mechanistically, truncated RNF43 variants trap CK1 at the plasma membrane, thereby preventing β-catenin turnover and propelling ligand-independent target gene transcription. Gene editing of human colon stem cells shows that RNF43 truncations cooperate with p53 loss to drive a niche-independent program for self-renewal and proliferation. Moreover, these RNF43 variants confer decreased sensitivity to anti-Wnt-based therapy. Our data demonstrate the relevance of studying patient-derived mutations for understanding disease mechanisms and improved applications of precision medicine.

**Keywords** cancer mutations; human colon organoids; PORCN inhibitors; RNF43; Wnt signaling

**Subject Categories** Cancer; Signal Transduction

The EMBO Journal (2020) 39: e103932

## Introduction

Aberrant activation of Wnt/β-catenin signaling is a key oncogenic event that confers an undifferentiated state and allows cancer cells to thrive outside their native niche constraint (Fujii *et al*, 2016; Nusse & Clevers, 2017; Zhan *et al*, 2017). In adult stem cells, Wnt signaling is curbed by the negative feedback regulators RNF43 and ZNRF3, two homologous transmembrane ubiquitin ligases that induce removal of the Wnt receptors FZD and LRP6 from the cell surface via ubiquitin-mediated endocytosis and lysosomal degradation (Hao *et al*, 2012; Koo *et al*, 2012). Within the stem cell niche, the activity of RNF43/ZNRF3 is counterbalanced by secreted proteins of the R-spondin (Rspo) family that form a complex with Leucine-rich repeat-containing G-protein-coupled receptor 4/5 (Lgr4/5) to mediate membrane clearance of RNF43/ZNRF3, promote Wnt receptor stabilization, and enhance Wnt responsiveness of stem cell populations (Carmon *et al*, 2011; de Lau *et al*, 2011; Chen *et al*, 2013; Peng *et al*, 2013a,b; Zebisch *et al*, 2013; Kabiri *et al*, 2014; Zebisch & Jones, 2015).

Mutational loss of *RNF43* and/or *ZNRF3* is observed in human malignancies of the colon, pancreas, stomach, ovary, endometrium, and liver (Furukawa *et al*, 2011; Wu *et al*, 2011; Ong *et al*, 2012; Jiang *et al*, 2013; Ryland *et al*, 2013; Zou *et al*, 2013; Giannakis *et al*, 2014). Inactivation of RNF43/ZNRF3-mediated feedback leads to an increased abundance of Wnt receptors at the cell surface, which renders cells hypersensitive to Wnt ligands in their environment (Koo *et al*, 2012). The resulting Wnt-dependent growth state drives tumorigenesis and generates a druggable addiction to Wnt ligands in these cancer subsets (Wu *et al*, 2011; Koo *et al*, 2012; Jiang *et al*, 2013; Ryland *et al*, 2013; Zhou *et al*, 2013; Lannagan

1 Department of Cell Biology and Oncode Institute, Center for Molecular Medicine, University Medical Center Utrecht, Utrecht, The Netherlands
2 Department of Experimental Biology, Faculty of Science, Masaryk University, Brno, Czech Republic
3 Department of Molecular Biology and Oncode Institute, Faculty of Science, Radboud Institute for Molecular Life Sciences, Radboud University Nijmegen, Nijmegen, The Netherlands
4 Central European Institute of Technology, Masaryk University, Brno, Czech Republic
5 Hubrecht Organoid Technology, Utrecht, The Netherlands
6 Laboratory for Experimental Oncology and Radiobiology and Oncode Institute, Center for Experimental and Molecular Medicine, Amsterdam UMC, Cancer Center Amsterdam, University of Amsterdam, Amsterdam, The Netherlands
7 Institute of Molecular Biotechnology of the Austrian Academy of Sciences (IMBA), Vienna BioCenter (VBC), Vienna, Austria
*Corresponding author. Tel: +31 88 75 57574; E-mail: m.m.maurice@umcutrecht.nl
†These authors contributed equally to this work

*et al*, 2019). Indeed, blocking Wnt ligand biogenesis by small-molecule inhibitors of the O-acyltransferase Porcupine (PORCN) suppresses the growth of RNF43-deleted pancreatic and small intestinal tumors in preclinical models (Chen *et al*, 2009; Jiang *et al*, 2013; Koo *et al*, 2015). This vulnerability has offered an opportunity to treat human cancers that are genetically defined by *RNF43* mutations with PORCN inhibitors. Currently, five small-molecule PORCN inhibitors are evaluated in clinical trials for cancer treatment (ClinicalTrials.gov, NCT01351103, NCT02278133, NCT02521844, NCT03447470, NCT02675946, NCT03901950, and NCT03507998).

Clearly, mutational inactivation of RNF43 is a prerequisite for Wnt-dependent growth (Koo *et al*, 2015). Recent cancer genome sequencing efforts, however, revealed a large diversity of genetic lesions within the *RNF43* locus of various human cancer types (www.cBioportal.org; Forbes *et al*, 2015). This mutational variability poses a challenge to unambiguously distinguish driver from passenger mutations and predict which mutations truly generate a Wnt-dependent growth state that can be exploited by targeted treatment. Insight in the mechanisms by which individual *RNF43* mutations contribute to cancer development and progression therefore is vital for the understanding of patient-specific disease mechanisms and the development of precision oncology strategies.

Here, we uncover a class of *RNF43* truncating mutations that drive inappropriate Wnt pathway activation by a mechanism distinct from *RNF43* Loss Of Function (LOF) mutations. Through capturing Casein kinase 1 (CK1) at the plasma membrane, these RNF43 mutants interfere with the turnover of the transcriptional coactivator β-catenin, promoting the transcriptional activation of Wnt target genes. When introduced in primary human colon stem cells, truncated RNF43 mutants induce a state of oncogenic stress and require prior inactivation of *TP53* to drive a niche-independent program for self-renewal and proliferation. Importantly, expression of oncogenic *RNF43* mutations, unlike conventional LOF *RNF43* mutations, reduces the potency of anti-Wnt-based therapy. Our results reveal the functional heterogeneity of cancer driver mutations in a single gene and demonstrate the importance of examining patient-derived mutations to uncover disease mechanisms, allow for improved patient stratification and applications of targeted therapy.

# Results

## Loss of the C-terminus endows the tumor suppressor RNF43 with oncogenic properties

RNF43 comprises a single-span transmembrane E3 ubiquitin ligase of 783 amino acids (Fig 1A). Binding and ubiquitination of Wnt receptors map to the N-terminal half of the RNF43 protein, including the extracellular (ECD), transmembrane (TM), and RING domains. These domains are followed by an extended C-terminal tail that contains conserved Ser-, His-, and Pro-rich regions to which no role has been assigned (Fig 1A). Notably, a third of reported *RNF43* cancer variants comprise nonsense or frameshift mutations that prospectively yield expression of C-terminally shortened RNF43 proteins for which functional consequences remain unknown (www.cBioportal.org; Giannakis *et al*, 2014; Forbes *et al*, 2015). To address this issue, we expressed RNF43 cancer variants carrying incremental C-terminal truncations in HEK293T cells and monitored

their impact on Wnt-induced β-catenin-mediated transcription. Strikingly, RNF43 variants truncated between residues K514-Q563 strongly induced β-catenin-mediated transcription, independent of supplementation with Wnt (Fig 1B). Unlike a well-defined LOF missense variant (I48T; Tsukiyama *et al*, 2015), these RNF43 mutants fully retained their ability to downregulate FZD receptors while strongly increasing cytosolic β-catenin levels (Figs 1C–F and EV1A) indicating a gain of oncogenic activity. Wnt pathway activation by the representative oncogenic variant RNF43 R519X remained unaffected by PORCN inhibitors, while these compounds fully suppressed Wnt signaling activity induced by the LOF cancer mutant I48T or by *RNF43/ZNRF3* deletion (Figs 1G and EV1B) (Jiang *et al*, 2013). Thus, in contrast to LOF mutations, RNF43 R519X induces pathway activation in a ligand-independent manner. Furthermore, RNF43 R519X strongly induced Wnt pathway activation in an *RNF43/ZNRF3*-knockout background, indicating that these mutants do not operate via a dominant-negative mechanism (Fig EV1B). Hence, truncated RNF43 variants gain competence to drive basal Wnt signaling and are functionally distinct from classical LOF mutations.

## Premature termination codons within the oncogenic region of *RNF43* avoid nonsense-mediated decay

More precise mapping of the oncogenic region using designed RNF43 truncations revealed that truncations located within D504-Q563 unleash β-catenin-mediated transcription, indicating that oncogenic activity requires retention of the Ser-rich region and loss of the Pro-rich region (Figs 1B and EV1C). Mutations introducing premature termination codons (PTC) within this *RNF43* region occurred in various cancer types, including pancreas, endometrium, ovarium, and colon (Appendix Table S1). Expression of inappropriately truncated proteins is commonly limited due to nonsense-mediated decay mRNA surveillance pathways (Lykke-Andersen & Jensen, 2015; Lindeboom *et al*, 2016). To investigate this issue, we employed CRISPR/Cas9 to introduce biallelic *RNF43* PTCs in SW480 APC-mutant colorectal cancer cells, in which *RNF43* is actively transcribed (Fig EV2A and B). Mutant *RNF43* mRNAs (V520fs/D516fs) were expressed even at increased abundance compared with parental cells (Fig EV2C and D), indicating that these transcripts are stable.

## Truncated RNF43 cancer variants interfere with downstream Wnt signaling events

Next, we aimed to identify the molecular requirements for the oncogenic activity of truncated RNF43 variants. Conventional wild-type RNF43 tumor suppressor activity relies on the RING-type E3 ligase domain that marks FZD for ubiquitin-mediated endocytosis and lysosomal turnover (Fig 1C and D; Hao *et al*, 2012; Koo *et al*, 2012). The introduction of RING domain-inactivating mutations still allowed for RNF43 R519X-mediated induction of basal β-catenin transcription, while responses to Wnt were further enhanced, likely due to the FZD stabilizing effects of this catalytically inactive RNF43 variant (Fig 2A; Koo *et al*, 2012). Furthermore, RNF43 R519X retained its ability to drive basal Wnt pathway activation when the ECD and TM domains were substituted by those of the unrelated transmembrane proteins CD16 and CD7 (Fig 2B and C). Similar

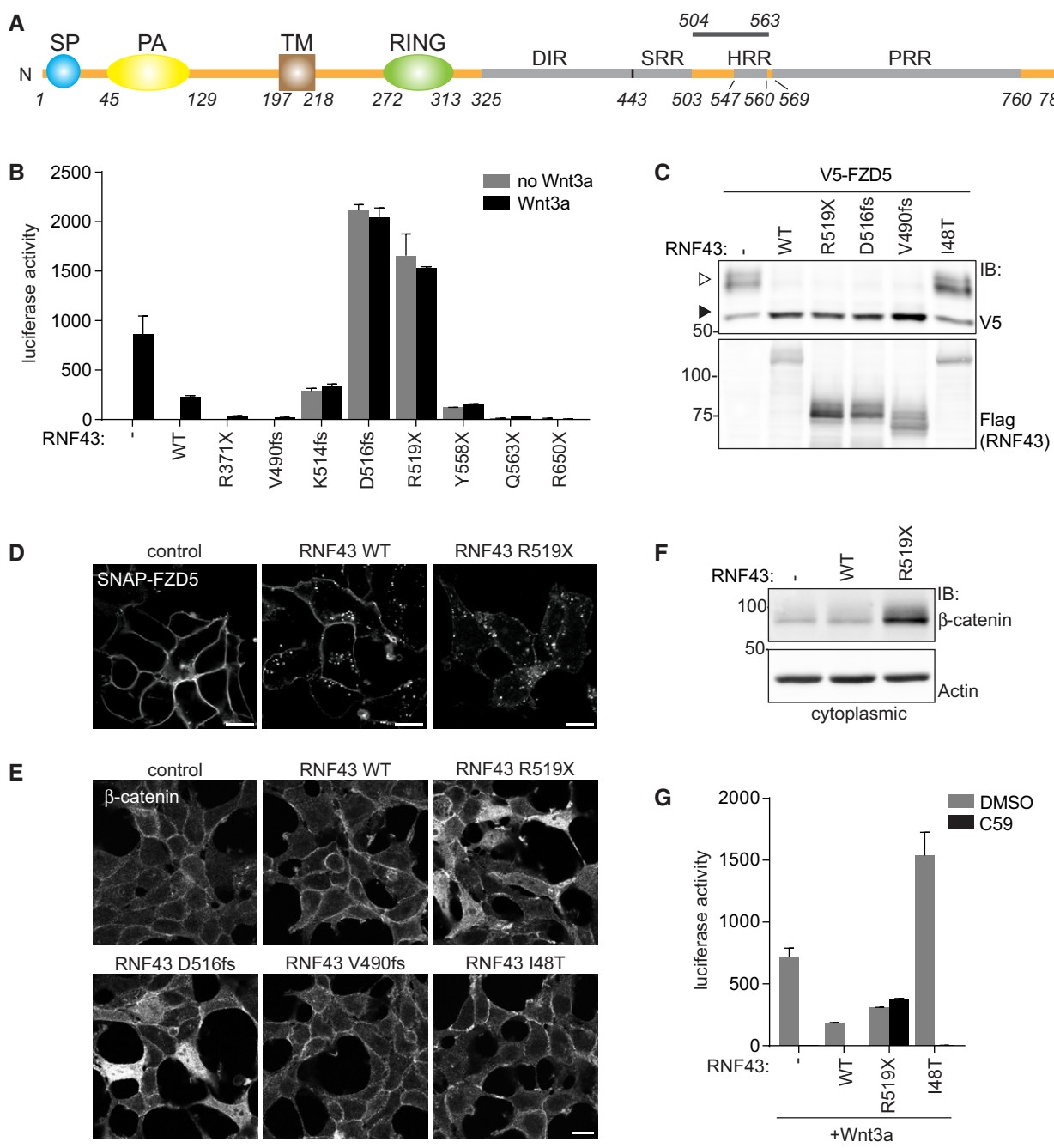

**Figure 1. Cancer truncations endow RNF43 with oncogenic properties.**

A  Schematic representation of the RNF43 protein. SP; signal peptide, PA; protease-associated domain, TM; transmembrane domain, RING; E3 ligase catalytic domain, DIR; Dishevelled-interaction region, SRR; Serine-rich region, HRR; Histidine-rich region, PRR; Proline-rich region. The region in which oncogenic truncations occur (aa 504–563) is indicated.

B  β-catenin-mediated reporter activity in HEK293T cells expressing the indicated RNF43 cancer mutants. Cells were treated with control medium (no Wnt3a) or Wnt3a-conditioned medium (Wnt3a) overnight. Average β-catenin-mediated reporter activities ± s.d. in *n* = 2 independent wells are shown.

C  Western blot analysis showing the effect of RNF43 cancer mutants on V5-FZD5 expression in HEK293T cells. Open and closed arrows indicate mature (post-Golgi) and immature (ER-associated) FZD5, respectively.

D  Confocal microscopy of surface labeled SNAP-FZD5 in HEK293T cells upon expression of RNF43 WT or R519X. Cells were chased for 30 min. Scale bars represent 10 μm.

E  Confocal microscopy of β-catenin localization in HEK293T cells expressing the indicated RNF43 cancer mutants. Scale bars represent 10 μm.

F  Western blot analysis of RNF43 WT and R519X for cytoplasmic β-catenin levels.

G  β-catenin-mediated reporter activity in HEK293T cells expressing Wnt3a and WT RNF43, oncogenic RNF43 (R519X) or a LOF RNF43 variant (I48T) after o/n treatment with DMSO or the PORCN inhibitor C59 (1 μM).

Data information: IB; immunoblot, WT; wild-type.

replacements in full-length RNF43 resulted in failure to downregulate FZD5 and loss of Wnt inhibitory activity (Fig 2C; Jiang *et al*, 2015). Thus, the truncated RNF43 cytosolic tail is required and sufficient to drive oncogenic β-catenin-dependent transcription by a mechanism independent of the ECD and RING domains. In line with this observation, RNF43 R519X-induced β-catenin-mediated transcription was insensitive to expression of Dishevelled (Dvl)-1 DEP-C, a Dvl fragment that binds the FZD cytosolic domains and blocks Wnt-mediated receptor activity (Fig 2D and Appendix Fig S1B; Tauriello *et al*, 2012). By contrast, expression of a dominant-negative variant of TCF4 (ΔN-TCF4), a nuclear β-catenin-binding partner (van Noort & Clevers, 2002), inhibited β-catenin-mediated transcription induced by oncogenic RNF43 truncations (Fig 2E). We

conclude that truncated RNF43 cancer variants affect a molecular step positioned downstream of the Wnt receptors and upstream of β-catenin-mediated transcription.

### Truncated RNF43 variants retain CK1 at the plasma membrane to drive β-catenin-mediated transcription

We next investigated whether truncated RNF43 interferes with the β-catenin destruction complex, which provides a central point for Wnt pathway regulation and is commonly targeted by inactivating mutations in cancer (Polakis, 2012; Zhan *et al*, 2017). BioID proximity labeling revealed interactions of WT RNF43 with destruction complex members Axin1, CK1α, CK1ε, and APC (Fig 3A;

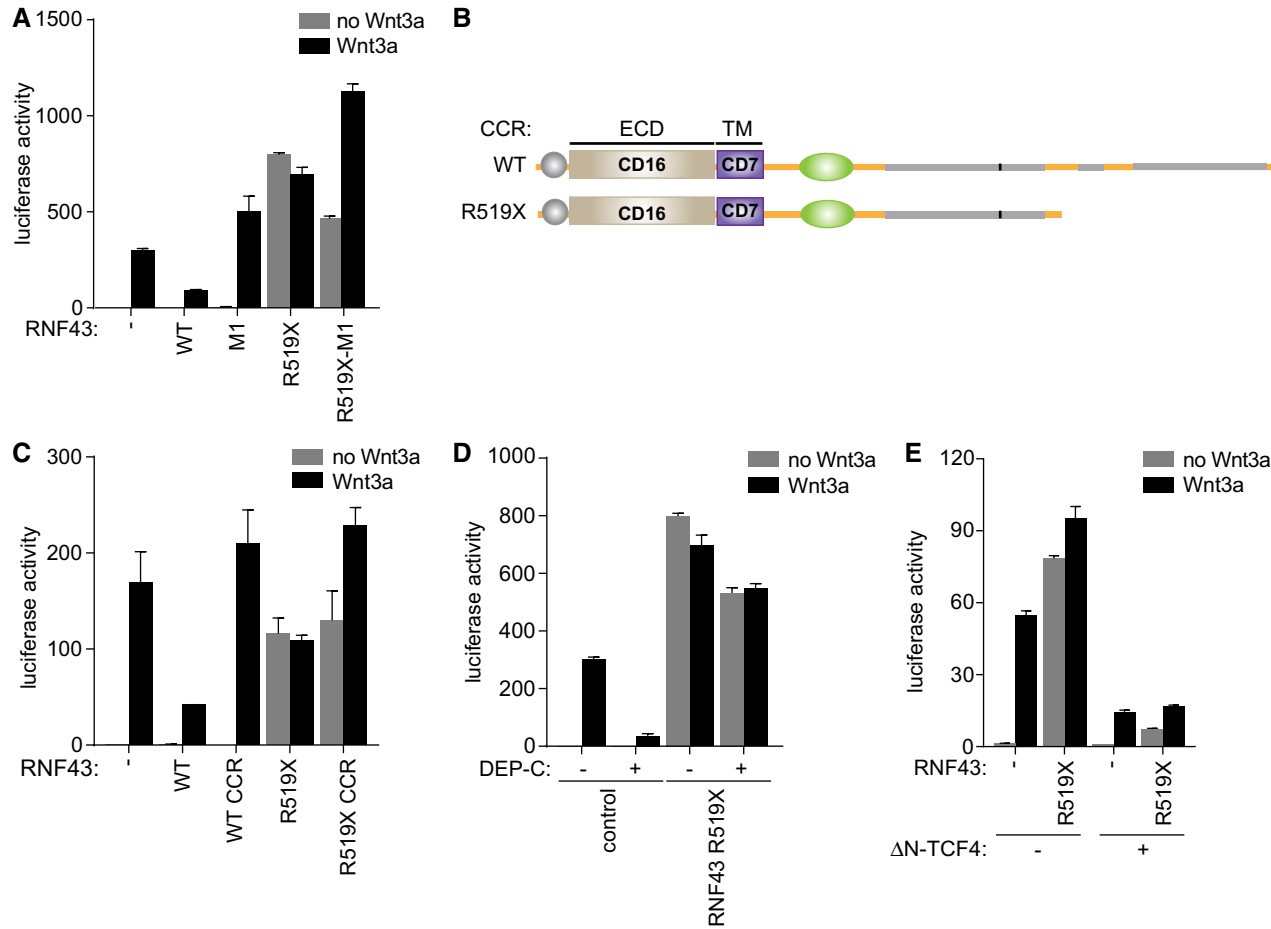

**Figure 2. Oncogenic RNF43 variants activate Wnt signaling downstream of the receptor complex and upstream of β-catenin.**

A β-catenin-mediated reporter activity in HEK293T cells expressing WT RNF43 and oncogenic RNF43 (R519X) harboring mutations C290S/H292S in the RING domain (M1). Cells were treated with control medium (no Wnt3a) or Wnt3a-conditioned medium (Wnt3a) overnight. Average β-catenin-mediated reporter activities ± s.d. in *n* = 2 independent wells are shown.

B Schematic of RNF43 constructs in which the extracellular and transmembrane domains (ECD and TM) are replaced by those of the unrelated CD16 and CD7, respectively. CCR; CD16-CD7-RNF43.

C β-catenin-mediated reporter activity in HEK293T cells expressing the indicated RNF43 constructs. Cells were treated with control medium (no Wnt3a) or Wnt3a-conditioned medium (Wnt3a) overnight. Average β-catenin-mediated reporter activities ± s.d. in *n* = 2 independent wells are shown.

D β-catenin-mediated reporter activity in HEK293T cells co-expressing oncogenic RNF43 (R519X) and the Dishevelled DEP-C tail. Cells were treated with control medium (no Wnt3a) or Wnt3a-conditioned medium (Wnt3a) overnight. Average β-catenin-mediated reporter activities ± s.d. in *n* = 2 independent wells are shown.

E β-catenin-mediated reporter activity in HEK293T cells co-expressing oncogenic RNF43 (R519X) and dominant-negative ΔN-TCF4. Cells were treated with control medium (no Wnt3a) or Wnt3a-conditioned medium (Wnt3a) overnight. Average β-catenin-mediated reporter activities ± s.d. in *n* = 2 independent wells are shown.

Data information: WT; wild-type.

Appendix Table S2). Strikingly, interactions of truncated oncogenic RNF43 variants with endogenous Axin1 and CK1α/ε were increased in comparison with RNF43 WT (Figs 3B and EV3A and B), while interactions with APC and GSK3β were not noticeably altered or even decreased (Figs 3B and EV3A, C and D). Moreover, expression of RNF43 R519X, but not WT or the non-oncogenic R371X variant, prompted a redistribution of Axin1, as well as endogenous CK1α and CK1ε, from cytosol to the plasma membrane (Figs 3C and D, EV3E and F). The interaction of CK1α/ε and RNF43 remained unaffected by depletion of Axin1 and its close homologue Axin2 (Fig EV3G), suggesting that CK1 interacts directly with the RNF43 cytosolic tail. Binding of CK1α/ε mapped to an intermittent region in the RNF43 C-terminus, composed of T483-Q488 and G492-S494 (Fig EV4A and B). Moreover, levels of CK1 binding correlated with Wnt pathway-activating ability of RNF43 R519X (Fig EV4B and C). These findings imply that retention of CK1 by RNF43-truncated cancer variants is essential for their oncogenic mode of action.

The observed increase in CK1 binding upon truncation suggests that downstream elements in the RNF43 tail normally regulate CK1 binding. As the RNF43 Q588X mutant performs normal suppressor activity (Fig EV4D), we evaluated a regulatory role of the Pro-rich region spanning Q563-Q588, located immediate adjacent to the oncogenic region. Indeed, the RNF43 ΔW564-P587 deletion variant displayed increased binding to CK1α/ε and, accordingly, was capable of driving oncogenic pathway activation (Fig EV4E and F). Thus, the W564-P587 sequence in the PRR normally regulates interactions of RNF43 with CK1, suggesting that these interactions are dynamic. Loss of this sequence is a requirement for enhanced CK1 binding and oncogenic activity.

### CK1-mediated phosphorylation of the truncated RNF43 cytosolic tail confers oncogenic Wnt pathway activation

To investigate whether RNF43 is a target for phosphorylation, we analyzed the phosphorylation status of RNF43 WT upon treatment with Wnt3a or Wnt3a/Rspo1. Upon Wnt3a treatment, overall Ser/Thr phosphorylation of RNF43 WT was reduced, while addition of Rspo1 increased phosphorylation status. These findings indicate that RNF43 phosphorylation is a regulated event during Wnt signaling (Appendix Table S3). In comparison with RNF43 WT, RNF43 R519X displayed an overall increase in basal phosphorylation, consistent with its ability to trap endogenous CK1 (Appendix Table S4). Furthermore, co-expression of CK1α, but not CK1ε, induced an increase in RNF43 WT phosphorylation, suggesting that CK1α is the preferred kinase for functional cooperation. Moreover, the majority of Rspo1-induced phospho-sites in RNF43 WT also became modified upon CK1α overexpression (Appendix Tables S3 and S4), suggesting that Rspo1 treatment might regulate CK1α activity toward RNF43.

We identified a non-canonical CK1 SLS target sequence at residues S500-S503 (SLSS) (Marin et al, 2003). Loss of this sequence (RNF43 S499X) abolished the ability of truncated RNF43 to induce β-catenin-mediated transcription (Fig EV1C), indicating it performs an essential role in driving oncogenic activity. In addition, Ala substitution of the SLSS motif (SLSS > ALAA) in full-length RNF43 abolished pathway suppression, while introduction of phospho-mimetic residues (SLSS > DLDD) promoted suppressor activity (Fig EV4G). In accordance, an RNF43 cancer variant carrying a

CK1-binding site deletion (ΔS486-G489>R; cBioportal) displayed LOF effects (Fig EV4G). The combined results thus indicate that RNF43 WT normally employs CK1 kinase activity to perform its tumor suppressor role. Importantly, the SLSS motif and CK1 binding are not required for FZD targeting, as both RNF43 ALAA and ΔS486-G489>R variants were still able to downregulate mature FZD5 similar to WT RNF43 (Fig EV4H).

Introduction of SLSS>ALAA or deletion of the CK1-binding site fully abrogated the capacity of RNF43 R519X to induce basal Wnt pathway activation, while SLSS>DLDD R519X mediated increased tumorigenic activity (Fig 3E). In line with its role as a CK1 target motif, mutation of the acidic region downstream of SLSS (residues 504–506) also affected the oncogenic activity of RNF43 R519X (Fig EV4C) (Marin et al, 2003). Thus, truncated RNF43 employs CK1 binding and phosphorylation to drive oncogenic Wnt pathway activation.

### Oncogenic *RNF43* mutations induce a *TP53*-dependent growth arrest in human colon organoids

Next, we investigated the impact of oncogenic RNF43 truncations on epithelial homeostasis, using human colon organoids (Jung et al, 2011; Sato et al, 2011). Introduction of CRISPR/Cas9-mediated frame shift mutations within the oncogenic region of the endogenous *RNF43* locus (onco-RNF43) yielded only a limited number of small organoid clones that failed to thrive, reminiscent of a senescent phenotype (Fig EV2A and Appendix Fig S2A; Ocadiz-Ruiz et al, 2017). Genotyping of a slowly expanding clone confirmed the presence of a mono-allelic onco-*RNF43* mutation (Appendix Fig S2B). This phenotype is strikingly different from *RNF43* LOF mutations that are well tolerated in intestinal organoids (Koo et al, 2012; Eto et al, 2018). We wondered how onco-RNF43 induces epithelial growth arrest. In line with our model of onco-RNF43-mediated CK1 sequestration, ablation of *Csnk1a1* (CK1α) from the mouse intestinal epithelium was shown previously to trigger massive Wnt pathway activation accompanied with p53-mediated cellular senescence (Elyada et al, 2011). Combined ablation of *Csnk1a1* and *Tp53* instigated formation of highly invasive carcinomas (Elyada et al, 2011). Similarly, we noted a co-occurrence of oncogenic *RNF43* frameshift mutations with mutations in *TP53* or senescence-associated genes in human cancer (Appendix Table S1), suggesting that *TP53* inactivation might be required to bypass an oncogenic stress-induced growth arrest. Indeed, combined onco-*RNF43*/*TP53*KO mutant clones rapidly appeared after CRISPR/Cas9 targeting, thrived in large numbers and allowed for the occurrence of biallelic onco-*RNF43* mutations (Appendix Fig S2A and B). Thus, loss of *TP53* creates a permissive cellular state for onco-RNF43 expression.

### Onco-RNF43 variants drive niche-independent growth in human colon organoids and confer decreased sensitivity to anti-Wnt-based therapy

A key feature of cancer pathway driver mutations is their ability to confer niche-independent growth, which is examined by depleting stem cell growth factors from the organoid culture medium (Sato et al, 2011; Fujii et al, 2016). We wondered if the ability of onco-RNF43 to drive basal β-catenin-mediated transcription alleviates the need for supplementation of colon organoids with Wnt and/or Rspo,

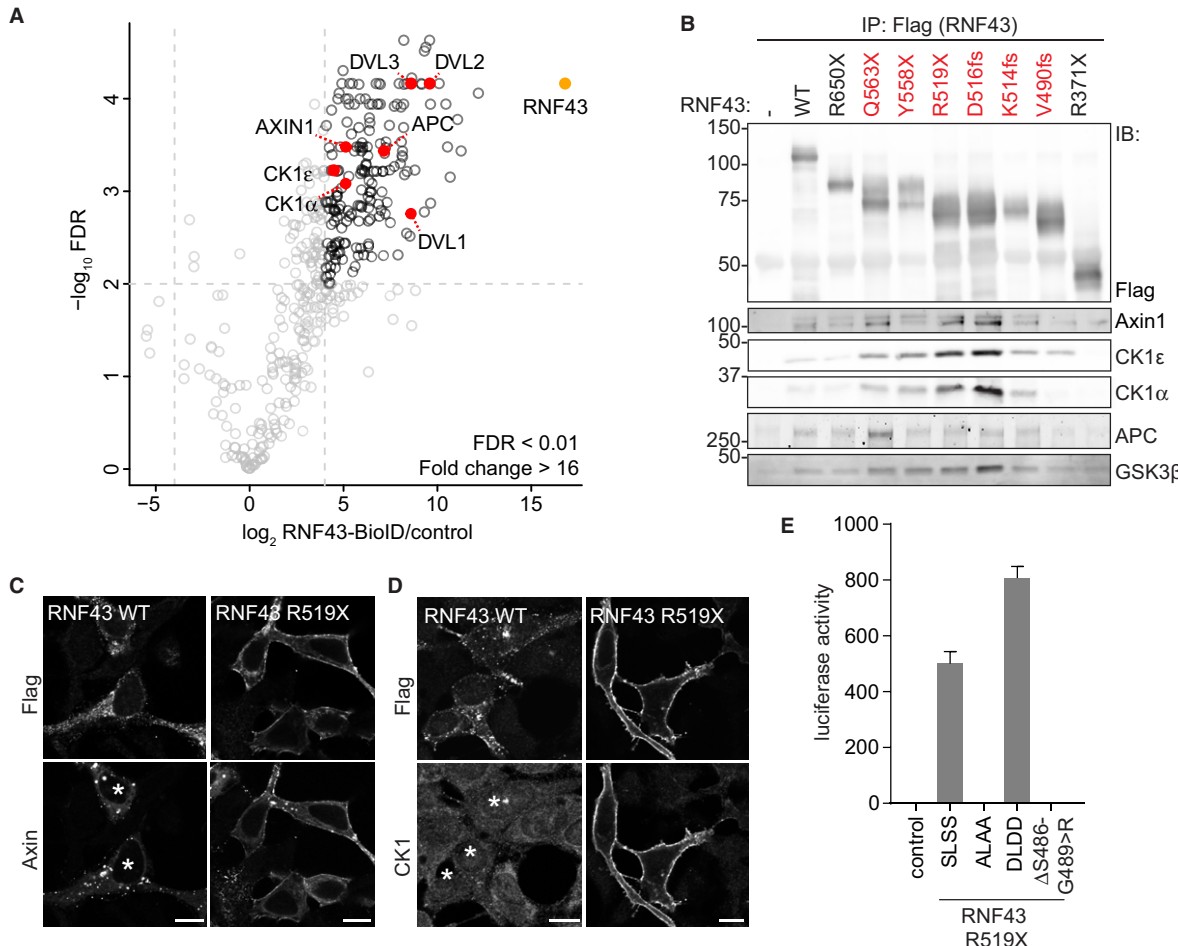

**Figure 3. Oncogenic RNF43 variants trap components of the β-catenin destruction complex at the plasma membrane.**

A Volcano plot showing proteins enriched after streptavidin pull-down of biotin-treated HEK293 cells stably expressing dox-inducible RNF43-BirA*. Striped line demarcates the empirical 0.01 False discovery rate (FDR) cutoff. Significantly enriched Wnt/β-catenin pathway components are highlighted in red and RNF43 in orange.

B Western blot analysis of endogenous destruction complex components that co-precipitated with the indicated RNF43 cancer variants expressed in HEK293T cells. Oncogenic RNF43 truncations are indicated in red.

C, D Confocal microscopy analysis of the subcellular localization of Axin1-GFP (C) and endogenous CK1α (D) upon expression of WT or oncogenic RNF43 (R519X). Scale bars represent 10 μm. RNF43 is visualized by Flag staining. Asterisks indicate RNF43-expressing cells.

E β-catenin-mediated reporter activity in HEK293T cells expressing non-mutated RNF43 R519X (SLSS) or the ALAA, DLDD, and ΔS486–G489>R mutants. Average β-catenin-mediated reporter activities ± s.d. in $n = 2$ independent wells are shown.

Data information: IP; immunoprecipitation, IB; immunoblot, WT; wild-type.

a Wnt-potentiating niche factor that induces membrane clearance of RNF43/ZNRF3 and allows for enhanced Wnt responsiveness of stem cell populations (Carmon *et al*, 2011; de Lau *et al*, 2011; Chen *et al*, 2013; Peng *et al*, 2013a,b; Zebisch *et al*, 2013; Zebisch & Jones, 2015). Omitting Wnt readily compromised viability of both WT and *TP53*KO organoid lines, as reported earlier (Sato *et al*, 2011; Yan *et al*, 2017), while onco-*RNF43/TP53*KO organoid growth remained largely unaffected (Fig 4A and Appendix Fig S3A). Although instant removal of Rspo was not tolerated by any of the organoid lines, onco-*RNF43/TP53*KO organoids displayed much greater tolerance to a step-wise decrease in Rspo concentrations when compared to WT and *TP53*KO organoid lines (Fig 4A and Appendix Fig S3A). We conclude that onco-*RNF43* mutations

confer decreased dependence on Wnt and Rspo niche factors, a hallmark of cancer cell growth.

To investigate the impact of onco-*RNF43* mutations on gene expression in colon epithelial cells, we performed RNA sequencing of WT, *TP53*KO, and onco-*RNF43/TP53*KO organoid lines grown in high Wnt/Rspo (20%) or no Wnt/low Rspo (0.2%) medium. Unsupervised clustering of significantly changing genes revealed four distinct clusters of gene expression dynamics (Fig EV5A). Onco-RNF43-mediated transcriptome alterations were markedly enhanced in no Wnt/low Rspo growth conditions (Fig EV5A and B). Therefore, we focused on 1448 genes that were differentially expressed in *TP53*KO versus onco-*RNF43/TP53*KO organoid lines grown in no Wnt/low Rspo (Fig EV5A and B). Of the 966 downregulated genes

in onco-*RNF43/TP53*KO organoids, a minor group of 123 genes was traced back to loss of *TP53*. The remaining 843 genes were specifically and consistently downregulated by onco-RNF43 expression.

Conversely, 482 genes were significantly increased in onco-RNF43-expressing organoids. Noticeably, this gene set was also upregulated in high Wnt/Rspo-treated organoids (Fig EV5A, right heatmap),

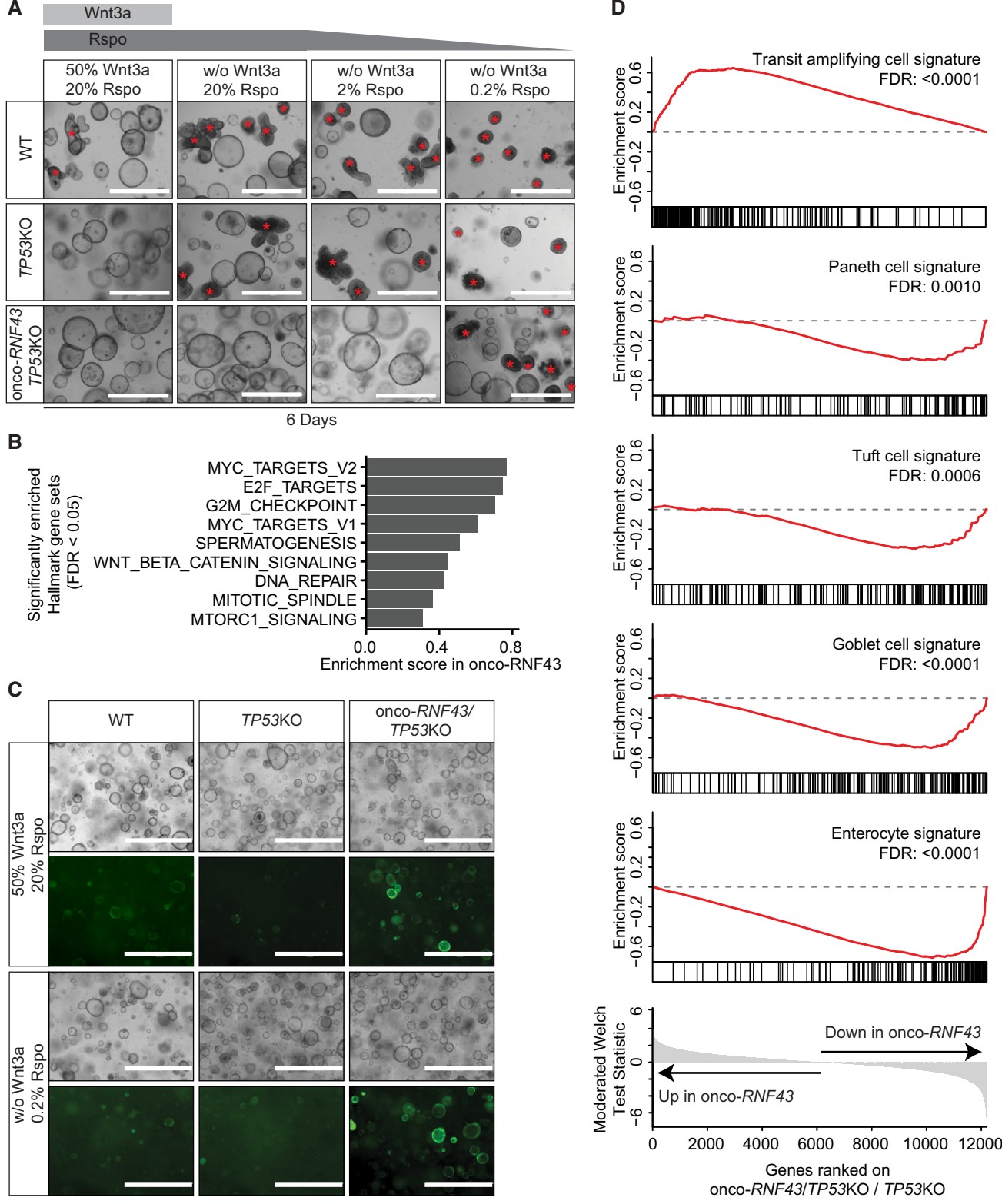

**Figure 4.**

◄

**Figure 4.  Onco-*RNF43/TP53*KO organoids display an oncogenic transcriptional profile that drives self-renewal and niche-independent growth.**

A   Bright-field microscopy images of WT, *TP53*KO, and onco-*RNF43/TP53*KO human colon organoid lines grown in medium with high Wnt/Rspo (20% conditioned medium (CM)) or without Wnt/Rspo (20, 2 or 0.2% CM). Images were taken 6 days after splitting. Scale bars represent 1,000 μm. Non-cystic, non-proliferative organoids are indicated with red asterisks.

B   Bar plot showing the enrichment scores of significantly enriched MSigDB hallmark gene sets in onco-*RNF43/TP53*KO compared to *TP53*KO organoids (FDR < 0.05).

C   Bright-field microscopy and fluorescence microscopy pictures of WT, *TP53*KO, and onco-*RNF43/TP53*KO human colon organoid lines grown in two different media and transduced with the TOP-GFP reporter. Images were taken 6 days after splitting. Scale bars represent 1,000 μm.

D   Gene Set Enrichment Analysis of onco-*RNF43/TP53*KO compared to *TP53*KO organoids in medium without Wnt/low Rspo (0.2%). Significantly changed intestinal cell-type gene sets from Haber *et al* (2017) are shown (FDR < 0.05).

indicating that onco-RNF43 confers a signature normally provided by the stem cell niche. Gene set enrichment analysis (GSEA) using MSigDB (Liberzon *et al*, 2015) revealed significant enrichment for Myc and E2F targets, Wnt signaling, DNA damage response and cell division by onco-RNF43 expression (Fig 4B). A lentiviral Wnt GFP reporter (Fuerer & Nusse, 2010) confirmed sustained Wnt signaling in onco-RNF43-expressing organoids in no Wnt/low Rspo growth conditions (Fig 4C, Appendix Fig S3B and C). Furthermore, differentiated cell type signatures (Haber *et al*, 2017) were lost in onco-RNF43-expressing organoids while profiles of transit-amplifying cells were notably enriched (Fig 4D). In summary, onco-RNF43 induces expression of a stem cell-like transcriptome in colonic epithelial cells and these effects are intensified in conditions where Wnt and Rspo are scarce.

The ability of onco-*RNF43* mutations to drive Wnt-*independent* signaling stands in stark contrast to the previously described role of *RNF43* LOF mutations that promote a Wnt-*dependent* growth state. Importantly, our findings predict differential sensitivity of these RNF43 mutational classes to treatment with PORCN inhibitors, Wnt antagonists that are currently evaluated for clinical treatment of *RNF43*-mutant cancer patients (Janku *et al*, 2015; Zhan *et al*, 2017; Rodon *et al*, 2018; Zhang & Lum, 2018). In line with retained Rspo-dependency, prolonged culturing in the presence of PORCN inhibitor C59 was not tolerated by any of the organoid lines. However, a large fraction of onco-*RNF43/TP53*KO organoid clones survived and recovered after 1 week of C59 treatment, while no viable clones were obtained for WT and *TP53*KO organoids (Fig 5A and B, and Appendix Fig S3D). These results indicate that onco-RNF43 expression reduces sensitivity to PORCN inhibitors. In line with these findings, onco-RNF43-expressing organoids accumulated a higher relative fraction of active, non-phosphorylated β-catenin under Wnt-depleted conditions, thus confirming compromised destruction complex activity (Appendix Fig S3E and F). Taken together, onco-RNF43 expression confers reduced sensitivity of human colon organoids to PORCN inhibitor treatment by promoting downstream β-catenin-mediated transcription.

## Discussion

Inappropriate Wnt/β-catenin signaling in cancer is achieved by two major mutational driver routes. Inactivation of the destruction complex is well studied and exemplified by the prominent driver role of *APC* mutations in colorectal cancer (Zhang & Shay, 2017). More recently, misregulation of Wnt receptor abundance emerged as an alternative oncogenic pathway (Jiang & Cong, 2016; Zhan *et al*, 2017). *RNF43* mutations, found in ~19% of colorectal cancer

cases, are mutually exclusive to *APC* mutations and considered a prime hallmark of Wnt-hypersensitive cancer subsets (Jiang *et al*, 2013; Giannakis *et al*, 2014; Koo *et al*, 2015). A primary and causal event for generation of a Wnt-hypersensitive state is the mutation-induced loss of RNF43/ZNRF3-mediated ubiquitination and endocytosis of Wnt receptors (Koo *et al*, 2015). As this Wnt receptor suppressor activity of RNF43 locates to its membrane-proximal regions, the functional relevance of truncating cancer mutations that remove more distal parts of the cytosolic tail thus far has remained unclear.

In this study, we uncover that the RNF43 C-terminus performs an additional tumor suppressor role, by regulating the activity of the downstream β-catenin destruction complex. We show that the cytosolic tail of RNF43 interacts with all core components of the destruction complex, in a highly regulated manner. We identify CK1 as a prominent interaction partner that binds the Ser-rich region and phosphorylates the RNF43 tail, which promotes RNF43-mediated tumor suppressor activity independent of FZD downregulation. A short sequence within the adjacent Pro-rich region inhibits CK1 binding, indicating that these interactions are normally dynamically regulated. Together, these findings lead to a model in which membrane-proximal parts of RNF43 act upon membrane-bound Wnt receptors, while more distant regions of the RNF43 tail are involved in dynamic regulatory interactions with components of the cytosolic destruction complex that are recruited to the receptors under Wnt-stimulated conditions (Fig 5C; Mao *et al*, 2001; Cliffe *et al*, 2003; Tamai *et al*, 2004; Zeng *et al*, 2005; Bilic *et al*, 2007; Li *et al*, 2012).

Importantly, our findings unveil a class of *RNF43* truncating cancer mutations that interfere with this second suppressor role to drive inappropriate Wnt pathway activation, by employing a mechanism distinct from that of *RNF43* LOF or *APC* mutations. Due to loss of the Pro-rich region, these mutants selectively acquire an increased binding capacity for both CK1 and Axin1. By a mechanism that involves trapping of CK1 at the membrane and hyperphosphorylation of the truncated tail, these RNF43 mutants interfere with β-catenin destruction complex activity in the cytosol, leading to the stabilization of non-phosphorylated β-catenin and the transcriptional activation of Wnt target genes (Fig 5C). Our data further reveal that loss of the RNF43 C-terminus is required and sufficient to fulfill this oncogenic activity, while both the extracellular domain and E3 ligase activity are dispensable.

In support of their oncogenic role, the activity of these onco-RNF43 variants does not require inactivation of the paralogue ZNRF3 (Koo *et al*, 2012). Different from *APC* or *RNF43* LOF mutations (Jiang *et al*, 2013; Giannakis *et al*, 2014; Drost *et al*, 2015), introduction of onco-*RNF43* mutations in human colon organoids induces a state of oncogenic stress and requires prior

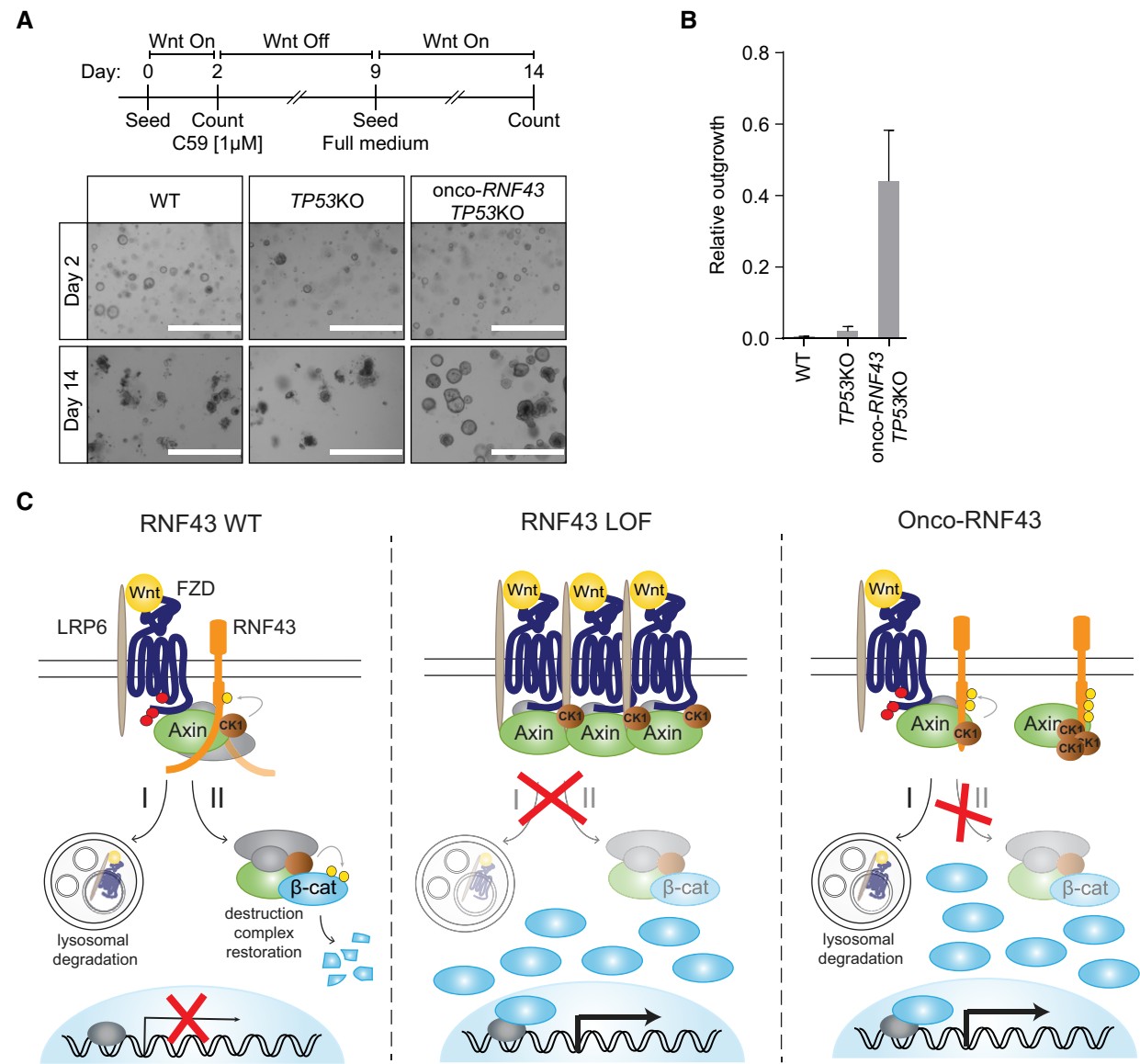

**Figure 5. Onco-RNF43 confers reduced sensitivity of colon organoids to Porcupine inhibitors.**

A  Top: Schematic experimental set-up of the clonogenic assay. Bottom: Bright-field microscopy images of WT, *TP53*KO, and onco-*RNF43/TP53*KO human colon organoid lines at days 2 and 14. Scale bars represent 1,000 μm.

B  Relative outgrowth of WT, *TP53*KO, and onco-*RNF43/TP53*KO organoid lines treated with the PORCN inhibitor C59 (1 μM) for 7 days. Graph shows the number of organoids at 5 days after splitting (day 14) relative to the number of organoids at day 2. Error bars represent ± s.d. of the mean of *n* = 3 experiments.

C  Model for mode of action RNF43 LOF and onco-RNF43 variants. (Left) RNF43 performs a bifunctional tumor suppressor role by (I) targeting Wnt receptors for endocytosis and lysosomal degradation, and (II) by transiently interacting with the destruction complex to reconstitute its activity in the cytosol and re-establish Wnt pathway inhibition. This second suppressor role involves CK1-mediated phosphorylation and an unknown molecular activity of the flexible cytosolic tail of RNF43. (Middle) Classical loss-of-function (LOF) mutations prevent RNF43 function at the plasma membrane, leading to Wnt receptor overexpression and, consequently, hypersensitivity of cancer cells to Wnt. (Right) Onco-RNF43 truncated variants are stably expressed at the plasma membrane and retain E3 ligase-dependent Wnt receptor downregulating activity. However, due to loss of the cytosolic tail, these mutants selectively trap CK1 and Axin1 at the plasma membrane, which prevents destruction complex assembly and drives uncontrolled β-catenin-mediated transcription of target genes.

Data information: WT; wild-type, LOF; loss-of-function, β-cat; β-catenin.

inactivation of *TP53* to exert their oncogenic role. Moreover, we show that onco-*RNF43* mutations inhibit differentiation, increase the number of stem cell progenitors, and drive a transcriptional program for proliferation, thus imposing a self-renewal phenotype onto cancer cells. Furthermore, these onco-*RNF43*-expressing organoids withstand treatment with PORCN inhibitors much better than WT or *TP53*KO organoids.

In summary, we identified a class of *RNF43* mutations that mediate a tumor suppressor-to-oncogene switch to drive downstream Wnt pathway activation. Unlike LOF mutations, the expression of

onco-RNF43 variants predicts prolonged survival in Wnt-depleted conditions and thus may decrease effectiveness of Wnt- or receptor-inhibiting treatment strategies. Our findings further imply that WT RNF43 performs a bifunctional tumor suppressor role, mediating ubiquitin-dependent Wnt receptor downregulation (Hao *et al*, 2012; Koo *et al*, 2012) as well as ubiquitin-independent regulation of destruction complex activity (Fig 5C). Our study demonstrates the importance of examining patient-derived mutations to identify novel tumorigenic molecular mechanisms, obtain a broader comprehension of signaling pathways in normal and cancer cells, and improve applications of precision cancer medicine.

# Materials and Methods

### Cell culture and transfection

Human Embryonic Kidney (HEK) 293 cells, HEK293T cells, and SW480 were cultured in RPMI or DMEM high glucose (Invitrogen), respectively, supplemented with 10% fetal bovine serum (GE Healthcare), 2 mM UltraGlutamine (Lonza), 100 units/ml penicillin, and 100 μg/ml streptomycin (Invitrogen). Cells were cultured at 37°C in 5% $CO_2$. Wnt3a-conditioned medium (CM) was obtained from L-cells stably expressing and secreting Wnt3a (Tauriello *et al*, 2010) cultured in DMEM low glucose (Invitrogen). Rspo- and Noggin-CM were produced as described before (Fenderico *et al*, 2019). For β-catenin-mediated reporter assays, HEK293T cells were stimulated overnight (o/n) and for protein expression experiments cells were stimulated for 3 h with Wnt3a-CM. Transfections were performed using either FuGENE 6 (Promega) according to manufacturer's protocol for β-catenin-mediated reporter assays and microscopy or polyethylenimine (PEI) for Western blot analysis. siRNA transfection was performed using Lipofectamine™ RNAiMAX (Thermo Fisher) according to manufacturer's protocol at 50 nM for 4 days. Control (#1) siRNA were obtained from Ambion (Thermo Fisher) and the Axin1 SMARTpool of siRNA from Dharmacon (Horizon). For co-transfection, plasmids were transfected 1 day before analysis.

### Plasmids and antibodies

Flag-Dishevelled1 DEP-C, TOPFlash, and FOPFlash luciferase reporter plasmids were described previously (Tauriello *et al*, 2010, 2012). Myc-β-catenin, ΔN (aa 1–32)-TCF4, and mouse Wnt3a were a kind gift of Hans Clevers (Hubrecht Institute, Utrecht, Netherlands). RNF43–2×Flag–HA and RING mutants were described previously (Koo *et al*, 2012). Plasmid for expression of human Axin1-GFP was described previously (Anvarian *et al*, 2016). Flag-APC-V5 was subcloned in pcDNA4 by PCR. All mutants were generated by either site-directed mutagenesis or by PCR-subcloning using Q5 High-Fidelity 2× Master Mix (NEB). All constructs were sequence verified. The following primary antibodies were used for immunoblotting (IB), immunofluorescence (IF), or immunoprecipitation (IP): goat anti-Axin1 (R&D systems), goat anti-CK1ε (Santa Cruz), goat anti-CK1α (Santa Cruz), rabbit anti-GSK3β (Cell Signaling), mouse anti-β-catenin (BD Transduction laboratories), mouse anti-Active-β-catenin (Millipore), rabbit anti-APC (Santa Cruz), rabbit anti-FLAG (Sigma-Aldrich), rat anti-HA (Roche), mouse

anti-HA (BioLegend), rabbit anti-V5 (Sigma-Aldrich), mouse anti-FLAG (M2; Sigma-Aldrich), mouse anti-V5 (Genscript), mouse anti-GFP (Roche), rabbit anti-TCF4/TCF7L2 (Cell Signaling), and mouse anti-Actin (MP Biomedicals). Primary antibodies were diluted conform manufacturer's instructions. Secondary antibodies used for IB or IF were used 1:8,000 or 1:300, respectively, and obtained from either Rockland or Invitrogen.

### β-catenin-mediated reporter assays

HEK293T cells were seeded in 24-well plates and transfected the next day with 30 ng of reporter construct TOPFlash or FOPFlash, 5 ng of thymidine kinase (TK)-Renilla and the indicated constructs. Cells were stimulated 6-h post-transfection with Wnt3a-CM o/n, then cells were lysed in Passive lysis buffer (Promega) for 20 min at room temperature (RT). IWP-2 (R&D systems) or C59 (Tocris) was used o/n at 5 or 1 μM respectively. Levels of Firefly and Renilla luciferase were measured using the dual-luciferase kit (Promega) accordingly to the manufacturer's instructions on a Berthold luminometer Centro LB960. Lysates were analyzed by Western blotting. For ΔN-TCF4, detection on Western blot was not feasible and instead expression is shown by microscopy (Appendix Fig S1A).

### Cell lysis and immunoprecipitation

HEK293T cells were transfected with PEI. 24 h post-transfection, cells were washed and collected in ice-cold PBS, and subsequently lysed in lysis buffer (50 mM Tris, pH 7.5, 150 mM NaCl, 0.5% Triton X-100, 5 mM EDTA, 1 mM DTT, 50 mM sodium fluoride, and protease inhibitors) for 30 min on ice followed by 30-min centrifugation at 16,100 $g$ at 4°C. Supernatants were used for immunoprecipitations (IPs) using 1 μg of the indicated antibody. Samples were left tumbling for 1 h at 4°C, followed by 1-h incubation with protein A or G beads (RepliGen and Millipore, respectively). For FLAG IPs, 15 μl pre-coupled FLAG M2 agarose (Sigma-Aldrich) was added for 1.5 h at 4°C while tumbling. Agarose beads were washed six times with lysis buffer and proteins were eluted in SDS sample buffer by 5 min boiling, or for FZD samples, incubated for 45 min at 37°C.

### Cell fractionation

24 h post-transfection, HEK293T cells were washed and collected in PBS and subsequently incubated for 10 min on ice in fractionation buffer (10 mM HEPES pH 7.9, 1.5 mM $MgCl_2$, 10 mM KCl, 1 mM DTT, and protease inhibitors) to allow the cells to swell. Cells were homogenized by 25–50 strokes in a Douncer after which the homogenization was visually evaluated by microscopy. The homogenate was centrifuged at 500 $g$ at 4°C for 10 min to obtain the nuclear pellet. The supernatant was subsequently centrifuged at 100,000 $g$ at 4°C for 1 h to separate the membrane fraction from the cytosolic fraction.

### Western blotting

Western blotting was performed using standard procedures with Immobilon-FL PVDF membranes (Millipore). In short: after protein transfer, the membranes were blocked for 1 h at RT in 1:1 ratio Odyssey blocking buffer (LI-COR): PBS. Primary antibodies were

incubated o/n at 4°C and secondary antibodies for 1 h at RT in the dark. The LI-COR Odyssey or Typhoon (GE Healthcare) infrared imaging systems were used for immunoblot analysis. Quantifications were performed using ImageQuant TL 8.2 (GE healthcare).

### Immunofluorescence and SNAP labeling

HEK293T cells were grown on laminin-coated glass coverslips in 24-well plates and transfected after 24 h. For SNAP labeling, cells were labeled with 1 µM SNAP-surface[549] (Bioke) for 15 min at RT, washed and subsequently chased for 30 min at 37°C. Cells were fixed in 4% paraformaldehyde or ice-cold methanol and blocked in PBS containing 2% BSA and 0.1% saponin. Primary and secondary antibody incubations were performed in blocking buffer for 45 min – 1 h at RT. Cells were mounted in ProLong Gold (Life Technologies) and analyzed using a Zeiss LSM510 or LSM700 confocal microscope.

### BioID for RNF43 interacting proteins

For the identification of RNF43-binding proteins, a previously described protocol was used (Roux *et al*, 2012). Briefly, pcDNA4-TO-RNF43-BirA*-HA was obtained by cloning RNF43 into pcDNA3.1 MCS-BirA(R118G)-HA (a gift from Kyle Roux; Addgene # 36047), which was subcloned into pcDNA4-TO. pcDNA4-TO-RNF43-BirA*-HA was transfected into T-REx™-293 cells and selected with 200 µg/ml of zeocin to obtain a stable RNF43 TetON -REx™-293 BioID cell line. Ten days after transfection, single cells were plated in 96-wells and grown in the presence of 100 µg/ml zeocin and 5 µg/ml of blasticidin. For validation, the selected clones were analyzed by Western blot and immunofluorescence for tetracycline (1 µg/ml, Santa Cruz Biotechnology) induced expression of RNF43-BirA*-HA fusion protein and enzyme activity of BirA* by biotin supplementation (50 µM, Santa Cruz Biotechnology). To identify RNF43 interacting proteins, TetON -REx™-293 BioID cells were seeded in 15cm dishes and were treated o/n with tetracycline and biotin. Non-induced cells were used as a negative control. Cells were lysed in 2 ml of lysis buffer containing 2% TX-100, 500 mM NaCl, 0.2% SDS, 50 mM Tris, pH 7.4 supplemented with protease inhibitors (Roche), phosphatase inhibitors (Calbiochem), 1 mM DTT, and PIC (Roche). Lysates were collected, sonicated, and cleared by centrifugation at 16,500 *g* for 15 min at 4°C. Streptavidin beads (Streptavidin Sepharose High Performance, GE Healthcare) were added to the lysates and incubated for 16-h tumbling at 4°C. Subsequently, beads were washed four times with lysis buffer and two times with 50 mM Tris, pH 7.4. Next, beads were washed with ammonium bicarbonate buffer and proteins were reduced with DTT and alkylated with iodoacetamide. Trypsin digestion on beads was performed o/n at 37°C. Resulting peptides were transferred into LC-MS vials and concentrated to 15 µl. LC-MS/MS analysis was performed on an RSLCnano coupled to an Orbitrap Elite system. MS/MS data processing was performed using Proteome Discoverer (version 1.4). Hits are classified as proteins with a minimum of two unique peptides present in at least two out of three replicates. Proteins were filtered using the cRAP contaminant database and proteins interacting with the BirA* tag or present in the negative control were subtracted. For volcano plot analysis, only proteins identified in at least all three replicates of either the control or the BioID sample were considered.

By using Perseus 1.5.5.3 (Tyanova *et al*, 2016) with default settings, missing values were imputed from a semi-random normal distribution around the lower detection limit of all detected proteins. False discovery rates (FDR) are calculated by a modified t-test (in Perseus) followed by Benjamini–Hochberg FDR adjustment.

### Identification of phosphorylation sites by mass spectrometry

Two methods were used for phosphorylation analysis by mass spectrometry. To analyze RNF43 phosphorylation upon Wnt3a and Rspo1 stimulation, HEK293T cells were seeded in 15-cm dishes and transfected at 80% of confluency with empty vector or RNF43–2×Flag–HA using PEI. 6 h after transfection half the medium was replaced with either control L-cell medium, Wnt3a-CM or 50% Wnt3a-CM/50% Rspo1-CM. After 20 h, cells were lyzed on ice for 30 min in 2 ml of lysis buffer containing 0.5% TX-100, 100 mM NaCl, 50 mM Tris, pH 7.5, 10% glycerol, 50 mM NaF freshly supplemented with 10 mM $Na_3VO_4$, 10 µM leupeptin, 10 µM aprotinin, and 1 mM PMSF and a phosphatase inhibitor cocktail (PhosSTOP, Sigma-Aldrich). Lysates were cleared by centrifugation at 16,100 *g* for 20 min at 4°C. Next, 45 µl of equilibrated M2-Flag beads (Sigma) was added and incubated 16-h tumbling at 4°C. Subsequently, beads were washed five times with lysis buffer and three times with Ammonium Bicarbonate 50 mM (pH 8). Bound proteins were then eluted off the beads first with 100 µl of 0.5% RapiGest SF (Waters), followed by 100 µl of 2.5% SDC; both dissolved in ammonium bicarbonate. Eluted proteins were reduced with 1 mM DTT and alkylated with 5.5 mM iodoacetamide. Samples were diluted threefold with ammonium bicarbonate before protein digestion. Proteins were first digested with endoproteinase Lys-C (Wako Chemicals) at 37°C for 2 h followed by trypsin (Promega) for 4 h and the tryptic peptides were subsequently digested with chymotrypsin (Roche) o/n. Protease digestion was stopped by addition of formic acid (FA) to a final concentration of 5% and any precipitates were removed by centrifugation. Peptides were desalted using an Oasis HLB 96-well plate (Waters). Phosphorylated peptides were enriched using Fe(III)-NTA cartridges (Agilent Technologies) in an automated fashion using the AssayMAP Bravo Platform (Agilent Technologies). Enriched samples were resuspended in 20 mM citric acid with 2% FA and analyzed with an UHPLC 1290 system (Agilent Technologies) coupled to an Orbitrap Q Exactive HF mass spectrometer (Thermo Scientific). The mass spectrometer was operated in data-dependent mode. Full-scan MS spectra from m/z 375–1,600 were acquired at a resolution and up to 12 most intense precursor ions were selected for HCD fragmentation. Raw files were processed using Proteome Discoverer (version 2.3.0.523). The database search was performed against the human Swissprot database using Mascot as search engine. Filtering was done at 1% false discovery rate (FDR) at the protein and peptide level. Only peptides with at least 6 amino acids and Mascot ion score above 20 were considered. Label-free quantification was performed using the Minora Feature Detector node. The phosphoRS feature was used for phosphorylation localization (minimum 98% site probability filter was applied). Quantified data were $log_{10}$-transformed and normalized.

For the analysis of CK1-mediated phosphorylation, a different protocol was used. HEK293 cells were seeded in 15-cm dishes and transfected at 80% of confluency with the indicated plasmids using PEI. Cells were lysed in 2 ml of lysis buffer containing 1% NP-40,

150 mM NaCl, 50 mM Tris, pH 7.5 supplemented with protease inhibitors (Roche), phosphatase inhibitors (Calbiochem), 1 mM DTT, and 10 mM NEM (N-ethylmaleimide) (Sigma-Aldrich). Lysates were collected, sonicated, and cleared by centrifugation at 16,100 g for 20 min at 4°C. Next, 2 μg HA-11 antibody (BioLegend) was added and samples were incubated for 1-h tumbling at 4°C. Then, 45 μl of equilibrated G-protein sepharose beads (GE Healthcare) was added to the sample and incubated 16-h tumbling at 4°C. Subsequently, beads were washed six times with lysis buffer, mixed with 50 μl of 2× Laemmli buffer, boiled for 5 min and loaded on 8% SDS-PAGE gels and separated. Gels were fixed with 50% methanol, 10% acetic acid, stained with 0.1% Coomassie brilliant blue (Sigma-Aldrich) in 20% methanol, 10% acetic acid for 2 h and destained using fixation solution. Next, corresponding 1-D bands were excised and processed for mass spectrometry analysis. Protein in gel pieces were alkylated, digested by trypsin, and subsequently cleaved by chymotrypsin. Digested peptides were extracted from gels. 1/10 of the peptide mixture was directly analyzed, and the rest of the sample was used for $TiO_2$ phosphopeptide enrichment. Both peptide mixtures were separately analyzed on LC-MS/MS system (RSLCnano connected to Orbitrap Elite; Thermo Fisher Scientific). MS data were acquired in a data-dependent strategy selecting up to top 10 precursors based on precursor abundance in the survey scan (350–2,000 m/z). High-resolution HCD MS/MS spectra were acquired in Orbitrap analyzer. The analysis of the mass spectrometric RAW data files was carried out using the Proteome Discoverer software (Thermo Fisher Scientific; version 1.4) with in-house Mascot (Matrix Science, London, UK; version 2.4.1) search engine utilization. The phosphoRS feature was used for phosphorylation localization and manually confirmed. Peptides with Mascot score > 20, rank 1 and with at least 6 amino acids were considered. Quantitative information assessment was done in Skyline software. Normalization of the data was performed using the set of phosphopeptide standards (added to the sample prior phosphoenrichment step; MS PhosphoMix 1, 2, 3 Light, Sigma) and by non-phosphorylated peptides identified in direct analyses. Clusters were analyzed by Orbitrap Script 2.0.

## smRNA FISH

SW480 cells were grown on coverslips for 24 h. For smFISH, samples were prepared as previously described (Lyubimova et al, 2013). Briefly, cells were fixed for 10 min with 4% Formaldehyde solution (Sigma-Aldrich) and 70% Ethanol o/n. Samples were then hybridized with Quasar 670 labeled RNF43 probes (Stellaris, Biosearch Technologies) and mounted to microscopy slides using Prolong Diamond Antifade (Invitrogen). Images were acquired using a deconvolution system (DeltaVision RT; Applied Precision) using 60× lens.

## gRNAs and genotyping

The pSpCas9(BB)-2A-Puro was obtained from Addgene (48139). gRNAs were generated as previously described (Ran et al, 2013). gRNA: Onco-*RNF43*- AGGCTGCATGTCCACTCGCT or TAGGGCTGCAGTACACTAGG; *RNF43* KO- ATTGCACAGGTACAGCGGGT; *ZNRF3* KO- GCCAAGCGAGCAGTACAGCG; *TP53* KO- GGCAGCTACGGTTTCCGTCT (a gift from Jarno Drost, PMC, Utrecht); AXIN2 KO-GCTTCCGTGAGGATGCCCCG. For genotyping, genomic DNA was isolated using QIAamp DNA micro kit (Qiagen). Primers for PCR

amplification using GoTaq Flexi DNA polymerase (Promega) were as follows: *RNF43*_Fw 5′-AGTGGATCTGGAGAAAGCTA-3′, *RNF43*_Rev 5′-ATTCAGCTGTAGTCTCCTCT-3′; *ZNRF3*_Fw 5′-TGATTACCATACAAGGTAGGTG-3′, *ZNRF3*_Rev 5′-CTCGTGCCTATAATTCCAGATA-3′; *TP53*_Fw 5′-CAGGAAGCCAAAGGGTGAAGA-3′, *TP53*_Rev 5′-CCCATCTACAGTCCCCCTTG-3′; *AXIN2*_Fw 5′-AGCTTTCCTTCCTCCGGTCTTC-3′, *AXIN2*_Rev 5′- GGTCACTACAGACTTTGGGGCT-3′. Products were cloned into the pGEM-T Easy vector system I (Promega) and subsequently sequenced using the T7 sequencing primer.

## Organoid culture

Healthy human colon tissue was isolated to establish human intestinal organoids for a previous study (Drost et al, 2015). Normal human colon organoids were cultured in advanced DMEM/F12 medium (Invitrogen), supplemented with B27 (Invitrogen), Nicotinamide (Sigma-Aldrich), N-acetylcysteine (Sigma-Aldrich), EGF (PeproTech), TGF-β type I receptor inhibitor A83-01 (Tocris), P38 inhibitor SB202190 (Sigma-Aldrich), Wnt3a-CM (50%), Noggin-CM (10%), and Rspo1-CM (20%) (full medium). Mutant *TP53* organoids were cultured in the presence of 5 μM Nutlin-3 (Cayman Chemical). Mutant RNF43 organoids were initially selected by withdrawing Wnt3a-CM and Rspo1-CM. Where indicated, the percentages of Wnt3a-CM and Rspo1-CM were adjusted. All experimentation using human organoids described herein was approved by the ethical committee at University Medical Center Utrecht (UMCU; TcBio #12-093). Informed consent for tissue collection, generation, storage, and use of the organoids was obtained from the patients at UMCU.

## Organoid electroporation

Organoid electroporation was performed as previously described (Fujii et al, 2015). Briefly, organoids were grown in 10 μM Y-27632 (Selleck chemicals) and 5 μM CHIR 99021 (Tocris) without CM for 2 days before electroporation. 24 h before electroporation, 1.25% DMSO was added to the culture medium. For electroporation, the NEPA21 electroporator was used with the configuration reported by Fujii et al (2015). 10 μg of pSpCas9(BB)-2A-Puro gRNA constructs were used to generate mutant lines. Organoids were recovered for 1 day by adding medium supplemented with 1.25% DMSO, 10 μM Y-27632, and 5 μM CHIR 99021 followed by another day with 10 μM Y-27632 and 5 μM CHIR 99021. 5-d post-electroporation organoids were grown in full medium. For CRISPR-engineered organoids, single clones were established by manual picking of individual organoids derived from single cells and genotyped. To visualize Wnt activity, organoids were transduced with 7xTcf-eGFP:SV40-PuroR (Top-GFP; a gift from Roel Nusse, Addgene #24305; Fuerer & Nusse, 2010) as described (Koo et al, 2011) and selected with Puromycin (Invivogen; 2 μg/ml).

## Clonogenic assay and quantification of organoids

Organoids were counted by eye based on their morphology. Proliferative organoids have a cystic morphology, where as differentiated and dead organoids are not cystic and are electron dense. Clonogenic assays were performed as previously described (Ramesh et al, 2018). Briefly, WT, *TP53*KO, and onco-*RNF43/TP53*KO organoid

lines were mechanically dissociated for 5 min by using a narrow Pasteur pipette. At day 2, organoids were counted and C59 (Tocris; 1 μM) was added for 7 days. At day 9, organoids were split and seeded in full medium and counted at day 14 ($n = 3$). Pictures were taken at day 2 and 14 using an Evos microscope. The relative outgrowth of organoids was based on the measurement of growth after 14 days corrected for the initial amount of organoids at day 2.

### RNA sequencing

Organoids were grown in full medium or medium with no Wnt/ 0.2% Rspo for 48 h and were lysed in RLT lysis buffer (Qiagen). RNA was obtained using the Qiagen QIAsymphony SP (Qiagen) according to the manufacturer's protocol. Single-end reads of 75 bp were aligned to GRCh37 with STAR (Dobin *et al*, 2013) with parameters outSJfilterIntronMaxVsReadN, chimJunctionOverhangMin, and chimSegmentMin set to 10,000,000, 15, and 15, respectively. Gene expression was quantified with edgeR with the human Ensembl transcript database version 74. Differential gene expression analysis was performed with DESeq2 (Love *et al*, 2014), and its 'rlogTransformation' function was used for variance stabilization of gene expression data. Clustering and generation of heatmaps were done with ComplexHeatmap (Gu *et al*, 2016). UpSet plots were created with the UpSetR R package (Lex *et al*, 2014). Intestinal cell type-specific gene sets for Gene Set Enrichment Analysis (GSEA; http://www.broad.mit.edu/gsea/) were defined as the 250 most specific cell-type signature genes per cell type from Extended fig table 3 of Haber *et al* (2017). Human orthologues of each cell-type gene set were determined with the biomaRt R package (Durinck *et al*, 2009). To calculate the false discovery rate of every GSEA, we performed 10,000 gene set perturbations. MYC targets V2 (version 2) was generated at more stringent inclusion criteria compared to version 1 and therefore is a smaller subset of MYC regulated genes.

### Statistical analysis

The statistical significance of interactions between identified proteins and RNF43-BirA* was determined by a Student's *t*-test by controlling for the false discovery rate with the Benjamini–Hochberg procedure. Significantly dynamic genes in the RNA-seq data were determined with the R package DESeq2 by using a Wald test. The false discovery rate of each Gene Set Enrichment Analysis was empirically determined by 10,000 gene set perturbations. Statistical details and sample numbers are specified in the figure legends.

## Data availability

The data that support the findings of this study are available from the corresponding author upon reasonable request. The RNA-seq data are publicly available at the NCBI GEO repository (accession number GSE129288; http://www.ncbi.nlm.nih.gov/geo/query/acc.cgi?acc=GSE129288). The mass spectrometry proteomics data for the BioID have been deposited to the ProteomeXchange Consortium via the PRIDE (Perez-Riverol *et al*, 2019) partner repository with the dataset identifier PXD020478.

**Expanded View** for this article is available online.

## Acknowledgements

We thank members of the laboratory of M.M.M. for discussions and suggestions. We thank Cara Jamieson for providing *RNF43/ZNRF3*-KO cells and Manja Omerzu and Lars Kemp for *AXIN2*-KO cells; Jarno Drost (Princess Máxima Centre for Paediatric Oncology, Utrecht, Netherlands) for *TP53* gRNA. This work is part of the Oncode Institute, which is partly financed by the Dutch Cancer Society. This work was supported by European Research Council Starting Grant 242958, the Netherlands Organization for Scientific Research NWO VICI Grant 91815604, ECHO Grant 711.013.012, and TOP Grant 91218050 (to M.M.M.), European Union Grant FP7 Marie Curie ITN 608180 "WntsApp" (to M.M.M.). V.B., T.R., and Z.Z. were supported by the Czech Science Foundation (project no. GX19-28347X). Projects CEITEC 2020 (LQ1601) and CIISB research infrastructure (LM2018127) from the Ministry of Education, Youth and Sports of the Czech Republic supported D.P. and K.H. who performed the LC-MS/MS measurements at the CEITEC Proteomics Core Facility. M.V. is supported by an ERC Consolidator Grant (771059).

## Author contributions

MS, NF, IJ, and MMM conceived and designed the experiments. MS, NF, IJ, TR, JMB, AC, LO, MO, EJ, KEB, KH, DP, and ZZ performed the experiments. MS, NF, IJ, TR, RGHL, JMB, AC, LO, MO, EJ, KEB, JPM, VB, B-KK, MV, and MMM analyzed the data. SFB provided essential reagents. MS, NF, IJ, TR, RGHL, VB, MV, and MMM wrote the manuscript, which was reviewed by all authors.

## Conflict of interest

The authors declare that they have no conflict of interest.

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
