## [Review Process File · The EMBO Journal]

RNF43 truncations trap CK1 to drive niche-independent self-renewal in cancer

Maureen Spit, Nicola Fenderico, Ingrid Jordens, Tomasz Radaszkiewicz, Rik G.H. Lindeboom, Jeroen M. Bugter, Alba Cristobal, Lars Ootes, Max van Osch, Eline Janssen, Kim E. Boonekamp, Katerina Hanakova, David Potesil, Zbynek Zdrahal, Sylvia F. Boj, Jan Paul Medema, Vitezslav Bryja, Bon-Kyoung Koo, Michiel Vermeulen, and Madelon M. Maurice

DOI: 10.15252/embj.2019103932

Corresponding author: Madelon Maurice (M.M.Maurice@umcutrecht.nl)

Review Timeline:

Submission Date:	11th Nov 19
Editorial Decision:	9th Jan 20
Revision Received:	4th May 20
Editorial Decision:	5th Jun 20
Editorial Correspondence:	16th Jun 20
Revision Received:	19th Jun 20
Accepted:	22nd Jun 20

Editor: Daniel Klimmeck

Transaction Report:

Dear Dr Maurice,

Thank you for the submission of your manuscript (EMBOJ-2019-103932) to The EMBO Journal. Please accept my sincere apologies for the unusual delay with the peer-review of your manuscript. Your manuscript has been initially sent to three reviewers, however one reviewer got much delayed and in the end did not send us his-her report after repeated chasers. We have received reports from the other two referees, which I enclose below, and decided to proceed with our decision based on these reports.

As you will see, the referees acknowledge the potential interest and novelty of your results, although they also express a number of major issues that will have to be conclusively addressed before they can be supportive of publication of your manuscript in The EMBO Journal. In more detail, referee #2 is concerned that the level of insights provided is not sufficient at this point and asks you to explore potential interplay between RNF43 and the destruction complex under normal conditions (pt 1,2) as well as to strengthen the evidence for endogenous relevance of the proposed CK1 titration model (pts. 4,5). Further, referee #1 requests additional experimentation on the control of RNF43 phosphorylation and specific relevance of the PRR domain (ref#1, pts. 1,2). In addition, the reviewers raise a number of points related to consistency between the results, additional controls required to corroborate the findings, overall data and method representation as well as wording, that would need to be conclusively addressed to achieve the level of robustness and clarity needed for The EMBO Journal.

I judge the comments of the referees to be generally reasonable and given their overall interest, we are in principle happy to invite you to revise your manuscript experimentally to address the referees' comments.

We do concur with the reviewers that providing conclusive support for an endogenous relevance of your observations will be important to move on with this work for publication in The EMBO Journal.

Please let me know any time if you have additional questions or need further input on the referee comments.

Please see below for additional instructions for preparing your revised manuscript.

Thank you for the opportunity to consider your work for publication. I again apologise for the delay and look forward to your revision.

Kind regards,

Daniel Klimmeck

Daniel Klimmeck, PhD
Editor

Before submitting your revision, primary datasets (and computer code, where appropriate) produced in this study need to be deposited in an appropriate public database (see <https://www.embopress.org/page/journal/14602075/authorguide#datadeposition>).

The accession numbers and database should be listed in a formal "Data Availability" section (placed after Materials & Method) that follows the model below (see also <https://www.embopress.org/page/journal/14602075/authorguide#availabilityofpublishedmaterial>). Please note that the Data Availability Section is restricted to new primary data that are part of this study.

Data availability

Our journal also encourages inclusion of *data citations in the reference list* to directly cite datasets that were re-used and obtained from public databases. Data citations in the article text are distinct from normal bibliographical citations and should directly link to the database records from which the data can be accessed. In the main text, data citations are formatted as follows: "Data ref: Smith et al, 2001" or "Data ref: NCBI Sequence Read Archive PRJNA342805, 2017". In the Reference list, data citations must be labeled with "[DATASET]". A data reference must provide the database name, accession number/identifiers and a resolvable link to the landing page from which the data can be accessed at the end of the reference. Further instructions are available at

- a point-by-point response to the referees' comments, with a detailed description of the changes made (as a word file).
- a word file of the manuscript text.
- individual production quality figure files (one file per figure)
- a complete author checklist, which you can download from our author guidelines (<http://emboj.embopress.org/authorguide>).
- Expanded View files (replacing Supplementary Information)

Further information is available in our Guide For Authors:

The revision must be submitted online within 90 days; please click on the link below to submit the revision online before 8th Apr 2020.

Link Not Available

Referee #1:

In this manuscript Spit and colleagues uncover that in addition to loss of function mutations in the tumor suppressor gene RNF43, a subset of mutations paradoxically functions as gain of function mutations and result in ligand-independent activation of the Wnt-bcatenin signaling pathway. These mutations drive niche-independent self-renewal of progenitor cells and likely participate in tumorigenesis. The authors delineated a trapping of the destruction complex CK1a as mechanism underlying the oncogenic activity of these RNF43 GOF mutations. In addition to the characterization of the importance of these mutations in cancer, the results presented suggests that RNF43 may have additional functions in controlling the activity of the destruction complex, in addition to regulate Frizzled receptor levels. The findings have important implications as Wnt pathway inhibitors are moving towards the clinic. Indeed, the authors show that the presence of RNF43 "oncogenic mutations" lead to ligand-independent signaling and resistance to PORC inhibitors (which are normally effective in the presence of early RNF43 truncation mutations).

This is a well conducted study and the data presented are convincing. The results are impactful in they reveal surprising functional consequences of cancer associated mutations that could have important clinical impact for patient stratification for Wnt inhibitors. The study also potentially uncovers a novel function of RNF43 for the regulation of destruction complex activity. I have suggestions below that could help strengthen the manuscript.

1. The main question I have is regarding the requirement of CK1a phosphorylation of RNF43 for its normal tumor suppressive function. The authors present evidence that mutation of the CK1a phosphorylation sites in RNF43 to alanine leads to loss of repressive function whereas mutations to aspartic acid leads to enhance repressive function. Is RNF43 phosphorylation regulated by Wnt ligand or Rspo or is this phosphorylation event constitutive?
2. Intriguingly the onco-mutation in RNF43 are restricted to truncations within 504-563 and requires maintaining the SRR and getting rid of the PRR domains. This may suggest a role for the PRR domain in negatively regulating binding/phosphorylation of RNF43 by CK1a, have the authors considered this possibility?
3. The authors show that one onco-mutant (R519X) leads to stabilization of beta-catenin while maintaining Frizzled degradation properties. It would be important to show the same effect for at least a second mutant.
4. At the top of page 4, the authors argue that the truncated transcripts are stable and that the encoded products promote transcription via a forward mechanism. I don't think evidence are presented to allow this conclusion. An alternative mechanism could be that the resulting mRNA transcript is more stable than the WT.
5. In Figure 2E, that ectopic expression of beta-catenin is unaffected by expression of the RNF43 onco-mutant is not unexpected. A better experiment would be to show that expression of a dominant negative LEF-TCF mutant would be effective in inhibiting bcatenin mediated transcription induced by the RNF43 mutant, whereas the dominant negative DEP-C Dvl mutant is ineffective. This would properly position the activity of the mutant upstream of the nuclear events. This is important given the recent report that RNF43 may have additional function downstream of bcatenin (PMID: 26350900).
6. In Figure 3B, the authors suggest that the interaction of RNF43 mutants and CK1a is increased while binding to APC and GSK3 is unchanged or decreased. It appears however that the binding to GSK3 is also increased, especially for D516x and D519x. Proper quantification may be needed.
7. In figure 3E, equal expression of the mutants needs to be shown to allow the authors to conclude their functional effects.
8. In figure 4A, it is unclear why the organoids containing the onco-RNF43 mutations are still reliant on Rspo whatsoever. They are of course much more resistant to the stepwise removal of Rspo but given the mechanism of action describing a total resistance to PORCN inhibitors, these results are harder to reconcile.
9. Throughout the text when referring the gene names, italics and caps should be used.
10. The authors refer to Wnt luciferase-reporter activity... They should rather refer to beta-catenin mediated or lef-tcf reporter activities...

Referee #2:

This manuscript reports an unexpected aspect of RNF43, which is an established E3 ligase for the Frizzled (FZD) family of Wnt receptors. RNF43 is a tumor suppressor gene product that is frequently mutated in cancers. The prevailing view has been that RNF43 promotes FZD

ubiquitination and degradation, thereby antagonizing Wnt signaling at the plasma membrane. RNF43 mutations in cancers are thus believed to compromise the E3 ligase action on FZD, thereby elevating the FZD protein level and Wnt signaling.

But the authors show that a common RNF43 cancer mutation, RNF43-R519X (a premature termination at R519), binds to components of the b-catenin-destruction complex, in particular to CK1a and CK1e. thereby titrating CK1s to the plasma membrane and leading to b-catenin stabilization. This action is downstream of FZD degradation and independent of Wnt stimulation. The authors refer to R519X as onco-RNF43.

In human colon organoids, CRISPR-mediated editing of the endogenous RNF43 gene with the R519X allele yielded only a mono-allelic mutation with little growth unless the P53 gene was also inactivated. In such a case the onco-RNF43 bi-allelic mutation was permitted. This Combined with p53 KO enabled some degree of Wnt/Rspo-independent expansion of colon organoids. This is consistent with the oncogenic nature of R519X mutation. RNA-seq was performed on these organoids, showing enhanced progenitor gene expression and reduced differentiation gene signatures. Further, these organoids exhibited less sensitivity to C59, a small molecule Porcupine inhibitor that blocks Wnt secretion and is being tested as potential cancer therapeutics. The authors thus suggest that Porcupine inhibitors may not be effective for cancers with R519X or similar mutations.

Overall the study is potentially interesting in uncovering a new class of RNF43 cancer mutations that appear to act in an unexpected manner, and has implications to our understanding of RNF43 in Wnt signaling and to therapeutic development. But given the unexpected nature of the findings that depart from the prevailing understanding of RNF43, there are some issues that require further and significant clarifications:

1. It is unclear how these findings are related to the normal function of RNF43. In the author's model (Fig. 5c), RNF43 has "a bifunctional tumor suppressor role by (I) targeting Wnt receptors for endocytosis and lysosomal degradation, and (II) by transiently interacting with the destruction complex to reconstitute its activity in the cytosol and re-establish Wnt pathway inhibition. This second suppressor role involves CK1-mediated phosphorylation and an unknown molecular activity of the flexible cytosolic tail of RNF43". But It seems that the point in "(II)" had been hardly addressed and remains quite confusing. Does RNF43 normally engages and regulates the destruction complex? If so does it promote the action of the destruction complex (the authors' view but there is no evidence) or does it have the opposite effect by competing for CK1a albeit perhaps weakly?

2. related, the authors identified a SLSS motif in RNF43 as potential phosphorylation sites of CK1, and generated ALAA, deletion (Delta 486-89), and DLDD in the R519X mutant (Fig. 3e) and the WT RNF43 (Fig. S2j). The data from these mutants are consistent with the SLSS motif being essential for RNF43 action. It is thus critical that the authors test these mutants in FZD degradation. If the SLSS motif is required for FZD degradation by RNF43, some interpretations may need to be modified accordingly and significantly. Also, the text referred to a delta 500-03 mutant. Is this the same as delta 486-89?

3. on the R519X oncogenic action: the authors' key point is that R519X titrates CK1a to the plasma membrane and away from the destruction complex. While some alanine mutants exhibited data consistent with this model, 501AAA and 504AAA mutants appear to be outliers despite that they bind to CK1 (Fig. S2h and i), posing questions to the model. The authors should address why these

two mutants behaved as outliers.

4. Related, the CK1 titration model, which is based on R519X overexpression, has additional challenges, in particular in terms of stoichiometry of RNF43-R519X versus CK1a. For example, if the R519X protein in a cell is much less abundant than CK1a, it would be difficult to envision how a titration scheme could work as proposed. Ideally The authors should measure/estimate the abundance of the endogenous R519X and CK1a proteins in the human colon organoids they generated (via quantitative mass spectrometry or immunoblotting).

5. Related, it would provide more support for the authors' model if they could show that b-catenin phosphorylation by CK1a is reduced by R519X independent of Wnt, ideally in human colon organoids without overexpression (i.e., mimicking a pathogenic condition).

6. can the authors quantify growth and the Wnt-reporter activity in colon organoids in Fig. 4? It is unsatisfactory to judge the selected images by eyes. In fact Quantifications were performed in Fig. 5a and 5b. how was this done?

Other issues:

7. Fig. 1b. please comment on why R371X , V490fs, and a few other mutants appear to be more potent than the WT?

8. Fig. 2d. The authors should show that the RNF43 M1 can be blocked by DEP-C.

9. Fig. 2e. this experiment does not say much, as R519X and b-catenin both activate TOP-Flash (i.e., one cannot perform epistasis using them).

10. Fig. 4b. what are Myc targets V1 vs V2?

11. Fig. S1d should be improved. It was difficult to follow the CRISPR-edited sequences. Were both alleles edited in exactly the same manner? Do SW480 cells have only 2 RNF43 alleles (many cancer lines are aneuploid)?

12. many figure legends are too brief and should be improved. It was difficult to follow what was being done in some of these figures. In general the writing should be improved given that this manuscript presents a convoluted/unexpected mechanism/story.

Point-by-point rebuttal**Reviewer #1:**

1. The main question I have is regarding the requirement of CK1 α phosphorylation of RNF43 for its normal tumor suppressive function. The authors present evidence that mutation of the CK1 α phosphorylation sites in RNF43 to alanine leads to loss of repressive function whereas mutations to aspartic acid leads to enhance repressive function. Is RNF43 phosphorylation regulated by Wnt ligand or Rspo or is this phosphorylation event constitutive?

We thank the reviewer for raising this interesting point. We analyzed phosphorylation of RNF43 upon treatment with Wnt3a or Wnt3a/Rspo1. As shown in **new Appendix Table S3**, RNF43 phosphorylation is reduced upon treatment with Wnt3a, while Rspo1 increases levels of RNF43 phosphorylation. Thus, these findings indeed reveal that phosphorylation of full-length RNF43 is a regulated event during signaling. In addition, we noted that the Rspo1-induced phospho-sites also become modified upon CK1 α overexpression (Appendix Table S4), suggesting that Rspo treatment might promote CK1 α activity towards RNF43. We now discuss these findings on page 6-7, line 198-209.

New Appendix Table S3. Phosphopeptides identified in RNF43 WT upon stimulation with Wnt3a and Rspo1.

Phosphosites	# P	RNF43 WT		
		control	Wnt3a	Wnt3a/ Rspo1
S251	1	5.50	5.00	5.68
S325	1	5.76	5.22	6.25
S443, S446	2	4.57	ND	5.60
S444	1	5.11	5.13	5.93
S532	1	5.88	5.16	5.79
S593	1	7.71	7.18	7.97
S603, S611	2	7.02	6.84	7.47
S611	1	6.30	5.95	6.74

The table summarizes the intensities of the identified phosphopeptides in the Log₁₀ scale. Areas of indicated phosphopeptides were normalized based on the median. Non-phosphorylated peptides were used for the second part of normalization. # P, number of phosphorylations observed on the peptide for the specific peptide spectrum match; ND, not detected

2. Intriguingly the onco-mutation in RNF43 are restricted to truncations within 504-563 and requires maintaining the SRR and getting rid of the PRR domains. This may suggest a role for the PRR domain in negatively regulating binding/phosphorylation of RNF43 by CK1 α , have the authors considered this possibility?

We thank the reviewer for this valuable suggestion. We investigated more precisely the role of the PRR which, based on the latest Uniprot classification, comprises amino acids 569-760. As the RNF43 Q588X mutant failed to induce basal Wnt pathway activation (**new Fig EV4D**), we deleted the region between Q563 (the boundary of the oncogenic region) and Q588, indicated as RNF43 Δ W564-P587. As shown in **new Fig EV4E**, this deletion of 23 aa unleashes the oncogenic activity of RNF43, promoting basal Wnt pathway activation similar to our previously shown onco-RNF43 variants. In addition, RNF43 Δ W564-P587 shows increased binding to CK1 (**new Fig EV4F**). Thus, this mutation shows that the PRR indeed is regulates the interaction with CK1 and RNF43 activity. These results are described on page 6, line 186-194.

New Fig EV4D-E. β -catenin-mediated reporter activity in HEK293T cells expressing the indicated RNF43 mutants in the presence and absence of Wnt3a. Average β -catenin-mediated reporter activities \pm s.d. in $n = 2$ independent wells are shown.

New Fig EV4F. Western blot analysis of endogenous CK1 α and CK1 ϵ co-precipitated with the indicated RNF43 variants expressed in HEK293T cells.

3. The authors show that one onco-mutant (R519X) leads to stabilization of betacatenin while maintaining Frizzled degradation properties. It would be important to show the same effect for at least a second mutant.

We have now included the same mutants as used for FZD5 degradation (Figure 1C) in the microscopy experiment (**new Fig 1E and Fig EV1A**). The D516fs truncation, similar to R519X, shows β -catenin stabilization. By contrast, the LOF mutant I48T and RNF43 V490fs (truncated outside oncogenic region) have no effect on the β -catenin levels and distribution.

New Fig 1E and EV1A. Confocal microscopy of β -catenin localization (1E) in HEK293T cells expressing the indicated RNF43 cancer mutants (EV1A). Scale bar represents 10 μ M.

4. At the top of page 4, the authors argue that the truncated transcripts are stable and that the encoded products promote transcription via a forward mechanism. I don't think evidence are presented to allow this conclusion. An alternative mechanism could be that the resulting mRNA transcript is more stable than the WT.

We agree with this comment, and have removed our statement on the potential feed forward mechanism (p5, line 138).

5. In Figure 2E, that ectopic expression of beta-catenin is unaffected by expression of the RNF43 onco-mutant is not unexpected. A better experiment would be to show that expression of a dominant negative LEF-TCF mutant would be effective in inhibiting bcatenin mediated transcription induced by the RNF43 mutant, whereas the dominant negative DEP-C Dvl mutant is ineffective. This would properly position the activity of the mutant upstream of the nuclear events. This is important given the recent report that RNF43 may have additional function downstream of bcatenin (PMID: 26350900).

As shown in **new Fig 2E**, Δ N-TCF4 expression indeed effectively inhibits onco-RNF43-mediated β -catenin transcription. This positions the activity of onco-RNF43 upstream of the nuclear events. We describe these results on page 5, line 157-160.

New Fig 2E. β -catenin-mediated reporter activity in HEK293T cells co-expressing oncogenic RNF43 (R519X) and dominant negative Δ N-TCF4. Average β -catenin-mediated reporter activities \pm s.d. in $n = 2$ independent wells are shown.

6. In Figure 3B, the authors suggest that the interaction of RNF43 mutants and CK1a is increased while binding to APC and GSK3 is unchanged or decreased. It appears however that the binding to GSK3 is also increased, especially for D516x and D519x. Proper quantification may be needed.

We have quantified the results of co-immunoprecipitation experiments (**new Fig EV3A**). The results confirm that the interactions of endogenous GSK3 β and APC with RNF43 are roughly equal for different onco-RNF43 variants. These findings are in line with the results of reverse IPs, in which interactions of onco-RNF43 with either of these components is not enhanced (Fig EV3C and EV3D). CK1 binding is strongly enhanced for onco-RNF43 variants, and despite some variation between experiments, Axin1 follows a similar pattern. Again, these findings are in line with the reverse IPs (Fig EV3B) and the microscopy data (Fig 4C and D).

New Fig EV3A. Quantification of the Western blot analyses of endogenous destruction complex components co-precipitating with the indicated RNF43 cancer variants shown in Fig 3B. Values were normalized to the amount of protein bound to RNF43 WT.

7. In figure 3E, equal expression of the mutants needs to be shown to allow the authors to conclude their functional effects.

The expression blots of all β -catenin-mediated reporter assays are included in Appendix Fig S4. Equal expression levels are observed for results shown in Fig 3E.

8. In figure 4A, it is unclear why the organoids containing the onco-RNF43 mutations are still reliant on Rspo whatsoever. They are of course much more resistant to the stepwise removal of Rspo but given the mechanism of action describing a total resistance to PORCN inhibitors, these results are harder to reconcile.

We apologize this aspect has not been clear. We do not claim a 'total' resistance of onco-RNF43 expressing organoids to PORCN inhibitors, but rather investigated a 'differential sensitivity' (page 9, line 296) of organoid lines. To do so, we treated organoids for 7 days with PORCN inhibitors and, after removal, evaluated the number of organoids that survived. The outcome showed that onco-RNF43/TP53KO organoids withstand treatment much better than WT or TP53KO organoids. Based on these findings, we conclude that onco-RNF43 expression confers resistance to PORCN treatment. Of note, prolonged treatment with PORCN inhibitors does not allow survival of onco-RNF43 organoids. The reasons for this are unclear. Nevertheless, our findings are relevant for clinical treatment with PORCN inhibitors as onco-RNF43-expressing cells clearly display a survival advantage under Wnt-depleted conditions as compared to WT cells.

9. Throughout the text when referring the gene names, italics and caps should be used.

We thank the reviewer for pointing this out and have corrected the gene names throughout the manuscript.

10. The authors refer to Wnt luciferase-reporter activity... They should rather refer to beta-catenin mediated or lef-tcf reporter activities...

We now employ ' β -catenin-mediated reporter activity' throughout the text.

Reviewer #2:

1. It is unclear how these findings are related to the normal function of RNF43. In the author's model (Fig. 5c), RNF43 has "a bifunctional tumor suppressor role by (I) targeting Wnt receptors for endocytosis and lysosomal degradation, and (II) by transiently interacting with the destruction complex to reconstitute its activity in the cytosol and re-establish Wnt pathway inhibition. This second suppressor role involves CK1-mediated phosphorylation and an unknown molecular activity of the flexible cytosolic tail of RNF43". But it seems that the point in "(II)" had been hardly addressed and remains quite confusing. Does RNF43 normally engage and regulate the destruction complex?

We apologize for the apparent lack of clarity on this point. Although the main focus of our study has been to uncover the mode of action of RNF43 truncated cancer variants, our findings clearly have implications for the understanding of the normal function of RNF43. A number of observations support the normal interaction of full-length RNF43 with the destruction complex: First, as a Wnt target gene, RNF43 operates under Wnt-stimulated conditions, in which the destruction complex localizes to the plasma membrane in complex with activated Wnt receptors (Stamos & Weis, 2013). Second, we have shown that full-length RNF43 engages endogenous destruction complex components by BioID/mass spec (Fig 3A) and by immunoprecipitation/Western blotting (Fig 3B). These findings were further confirmed using overexpressed proteins in co-IP experiments (Fig EV3C-D) and in microscopy (Fig 3C, 3C and EV3F). Third, as truncation enhances the interaction of RNF43 with CK1 and Axin, we anticipate that these interactions are normally subjected to regulation by C-terminal regions (Fig 3B, EV3B and EV4A). Of note, in response to reviewer 1, we now fine-mapped this regulatory activity of CK1 binding to a 23-aa motif in the Pro-rich region (PRR) (**new Fig EV4F**). Thus, these findings suggest that CK1 binding is normally transient and highly regulated. Fourth, we mapped the interaction of RNF43 with CK1 to a sequence located downstream of known domains involved in receptor downregulation (Fig EV4B). Together, these findings led to a model in which membrane-proximal parts of RNF43 act upon membrane-bound Wnt receptors, while more distant regions in the tail are involved in binding components of the cytosolic destruction complex.

New Fig EV4F. Western blot analysis of endogenous CK1α and CK1ε co-precipitated with the indicated RNF43 variants expressed in HEK293T cells.

If so does it promote the action of the destruction complex (the authors' view but there is no evidence) or does it have the opposite effect by competing for CK1α albeit perhaps weakly?

To shed light on this issue, we employed the RNF43 M1 mutant that lacks Ub ligase activity and fails to downregulate Wnt receptors, thus failing to perform activity "(I)" (Koo *et al*, 2012). We expressed RNF43 M1 in RNF43/ZNRF3-knockout cells to rule out dominant-negative effects of the mutant and evaluated its suppressing effects on the basal β-catenin activity present in these cells. As shown in **new Appendix Fig S1C**, the M1 variant is still capable to partially suppress β-catenin-mediated

transcription in *RNF43/ZNRF3*-knockout cells, in a dose-dependent manner. These findings thus confirm that full-length RNF43 carries β -catenin suppressor activity beyond Wnt receptor downregulation (activity "I"). Conversely, we now also show that mutants that lack activity "I" are still capable to perform FZD downregulation (**new Fig EV4H**; see our reply to point 2).

New Appendix Fig S1C. β -catenin-mediated reporter activity in HEK293T double knock out (dKO) *RNF43/ZNRF3* (R/Z) cells expressing increasing amounts of RNF43 WT or the ligase dead mutant (M1) in the presence and absence of Wnt3a. Average β -catenin-mediated reporter activities \pm s.d. in n = 2 independent wells are shown.

New Fig EV4H. Western blot analysis showing the effect of the indicated RNF43 variants on V5-FZD5 expression in HEK293T cells. Open and closed arrows indicate mature (post-Golgi) and immature (ER-associated) FZD5, respectively.

2. related, the authors identified a SLSS motif in RNF43 as potential phosphorylation sites of CK1, and generated ALAA, deletion (Delta 486-89), and DLDD in the R519X mutant (Fig. 3e) and the WT RNF43 (Fig. S2j). The data from these mutants are consistent with the SLSS motif being essential for RNF43 action. It is thus critical that the authors test these mutants in FZD degradation. If the SLSS motif is required for FZD degradation by RNF43, some interpretations may need to be modified accordingly and significantly.

We thank the reviewer for this valuable suggestion. The introduction of the SLSS>ALAA and delta486-89 mutations in full-length RNF43, which compromises suppressor activity, does not affect the capability of the protein to downregulate FZD (**new Fig EV4H**), further supporting a dual suppressor role of the protein.

Also, the text referred to a delta 500-03 mutant. Is this the same as delta 486-89?

We apologize for the confusion. Residues 500-503 refer to the SLSS CK1 target sequence, while the Δ 486-89 refers to the CK1 binding site. We have now clarified this issue on page 7, line 211-217.

3. on the R519X oncogenic action: the authors' key point is that R519X titrates CK1 α to the plasma membrane and away from the destruction complex. While some alanine mutants exhibited data consistent with this model, 501AAA and 504AAA mutants appear to be outliers despite that they bind to CK1 (Fig. S2h and i), posing questions to the model. The authors should address why these two mutants behaved as outliers.

The 501AAA mutation prevents phosphorylation of the essential SLS motif, and the 504AAA interferes with the acidic cluster downstream of this motif. Interference with the acidic cluster generally reduces or eliminates phosphorylation of the preceding SLS motif (Marin *et al*, 2003). Thus, while both mutants still interact with CK1, phosphorylation of their SLS motif is affected. Therefore, oncogenic activity is abrogated in these mutants (similar to the RNF43 ALAA mutant). We have now clarified this in the text (page 6, lines 220-222).

4. Related, the CK1 titration model, which is based on R519X overexpression, has additional challenges, in particular in terms of stoichiometry of RNF43-R519X versus CK1 α . For example, if the R519X protein in a cell is much less abundant than CK1 α , it would be difficult to envision how a titration scheme could work as proposed.

We fully agree it is unlikely that onco-RNF43 titrates the entire pool of CK1 α . As CK1 performs multiple tasks in the cell (Knippschild *et al*, 2014), its activity is directed by the complexes in which it is incorporated. Thus, only a fraction of the CK1 α pool is dedicated to β -catenin degradation. For comparison, only <10% of the total pool of GSK3 β has been estimated to take part in the β -catenin destruction complex (Patel & Woodgett, 2017). We anticipate that onco-RNF43 interacts with the destruction complex-bound fraction of CK1. Of note, our model of onco-RNF43-mediated pathway activation does not only rely on CK1 titration but also requires phosphorylation of RNF43 itself. We now adapted our statements in the text to clarify this point (page 10-11, line 335-344).

Ideally The authors should measure/estimate the abundance of the endogenous R519X and CK1 α proteins in the human colon organoids they generated (via quantitative mass spectrometry or immunoblotting).

Unfortunately, it is currently not technically feasible to quantify endogenous RNF43 protein levels. First, we are not able to detect endogenous RNF43 using commercial or homemade antibodies, and in any case a comparison of immunoblots performed with two different antibodies will not allow us to draw conclusions on relative protein levels. Second, we were not able to identify RNF43-derived peptides in colon organoids lysates using mass spec, despite multiple attempts. Nevertheless, we were able to address the consequences of endogenous onco-RNF43 expression on β -catenin phosphorylation in human colon organoids, as shown below in point 5.

5. Related, it would provide more support for the authors' model if they could show that b-catenin phosphorylation by CK1 α is reduced by R519X independent of Wnt, ideally in human colon organoids without overexpression (i.e., mimicking a pathogenic condition).

We thank the reviewer for this valuable suggestion. We now compared levels of endogenous, non-phosphorylated β -catenin in WT, TP53KO and onco-RNF43/TP53KO colon organoids in conditions of Wnt depletion (Wnt/Rspo withdrawal plus treatment with PORCN inhibitor). The results show that the relative fraction of non-phosphorylated β -catenin is increased in onco-RNF43/TP53KO human colon organoids, supporting a Wnt-independent decrease in destruction complex activity upon endogenous onco-RNF43 expression (**new Appendix Fig S3D and S3E**).

New Appendix Fig S3D and E. (D) Western blot analysis of endogenous β -catenin immunoprecipitated from WT, TP53KO and onco-RNF43/TP53KO human colon organoid lysates. Organoids were grown in full medium for 5 days and 24h before lysis placed in medium with no Wnt3a/no Rspo supplemented with PORCN inhibitor C59 (1 μ M). Levels of total β -catenin and active, non-phosphorylated β -catenin were detected. **(E)** Quantification of the Western blot presented in (D). Active, non-phosphorylated β -catenin levels were quantified and normalized to total levels of β -catenin. Graph represents the mean.

6. can the authors quantify growth and the Wnt-reporter activity in colon organoids in Fig. 4? It is unsatisfactory to judge the selected images by eyes.

We have now indicated differentiated and dead organoids in Fig 4A for clarification (**new Fig 4A**). We also quantified the fraction of surviving proliferative organoids that display a common cystic appearance (**new Appendix Fig S3A**).

New Fig 4A. Bright-field microscopy images of WT, TP53KO and onco-RNF43/TP53KO human colon organoid lines grown in medium with high Wnt/Rspo (20% conditioned medium (CM)) or without Wnt/Rspo (20, 2 or 0.2% CM). Images were taken 6 days after splitting. Scale bars represent 1000 μ m. Non-cystic, non-proliferative organoids are indicated with red asterisks.

New Appendix Fig S3A. Fraction of WT, TP53KO and onco-RNF43/TP53KO human colon organoids that presented a cystic, proliferative morphology in the conditions described in Fig 4A. Error bars represent 95% confidence interval (Wilson/Brown test). $n = 24 - 63$ organoids per condition.

To quantify Wnt reporter activity, we examined TOP-GFP levels by Western blotting (**new Appendix Fig S3B and S3C**). The results confirm that enhanced levels of TOP-GFP are present in onco-RNF43-expressing organoids, indicating increased levels of Wnt signaling.

new Appendix Fig 3B

new Appendix Fig 3B

New Appendix Fig S3B and C. (B) Western blot analysis of WT, TP53KO and onco-RNF43/TP53KO human colon organoid lines grown in two different media and transduced with the TOP-GFP reporter. Organoids were lysed 6 days after splitting and the indicated antibodies were used for detection. **(C)** Quantification of the Western blot presented in (B). GFP levels were quantified and normalized to actin. Graph represents the mean normalized to no Wnt3a/0.2% Rspo medium \pm s.d. of 2 independent experiments.

In fact Quantifications were performed in Fig. 5a and 5b. how was this done?

Quantifications for Figure 5A and B were performed as described above, with normalization of the number of cystic organoids that grew out (on day 14) to the number of organoids that were seeded (day 2). We have adjusted the Material and Methods section and figure legend to clarify this issue.

7. Fig. 1b. please comment on why R371X, V490fs, and a few other mutants appear to be more potent than the WT?

Indeed, we noticed that these truncations appear more potent in mediating Wnt signaling inhibition. We do not know the molecular basis for this, as the main focus of our study has been on the activation of signaling by cancer truncations. Our preliminary data show that these RNF43 truncations display enhanced interaction with FZD5, potentially explaining their increase in potency. While we agree this is an interesting observation, addressing the underlying molecular activities would be beyond the scope of the current project.

8. Fig. 2d. The authors should show that the RNF43 M1 can be blocked by DEP-C.

We performed the suggested experiment. The results show that the dominant-negative effect of RNF43 M1 can be blocked by DEP-C expression (**new Appendix Fig S1B**).

New Appendix Fig S1B

New Appendix Fig S1B. β -catenin-mediated reporter activity in HEK293T cells co-expressing Dvl-DEPC together with RNF43 WT or the ligase dead mutant (M1) in the presence and absence of Wnt3a. Average β -catenin-mediated reporter activities \pm s.d. in $n = 2$ independent wells are shown.

9. Fig. 2e. this experiment does not say much, as R519X and b-catenin both activate TOP-Flash (i.e., one cannot perform epistasis using them).

We thank the reviewer for this comment. As suggested by reviewer 1 (#5), we have now replaced this experiment by overexpression dominant negative TCF4 (Δ N-TCF4). The results show that Δ N-TCF4 effectively inhibits onco-RNF43-induced β -catenin-mediated transcription (**new Fig 2E**), positioning the activity of onco-RNF43 upstream of the nuclear events. We describe these results on page 5, line 157-160.

New Fig 2E. β -catenin-mediated reporter activity in HEK293T cells co-expressing oncogenic RNF43 (R519X) and dominant negative Δ N-TCF4. Average β -catenin-mediated reporter activities \pm s.d. in $n = 2$ independent wells are shown.

10. Fig. 4b. what are Myc targets V1 vs V2?

MYC_TARGETS_V1 and MYC_TARGETS_V2 are 'hallmark' gene sets of genes regulated by MYC, which are manually curated by the Molecular Signatures Database (MSigDB). The moderators of MSigDB generated and validated these gene sets based on three MYC overexpression studies, two MYC knock down studies, and three studies comparing MYC-high vs. MYC-low cells. MYC targets V2 (version 2) is a smaller subset of MYC regulated genes compared to version 1, because it was generated at more stringent inclusion criteria. We now clarified this in the Material and Methods section.

11. Fig. S1d should be improved. It was difficult to follow the CRISPR-edited sequences. Were both alleles edited in exactly the same manner? Do SW480 cells have only 2 RNF43 alleles (many cancer lines are aneuploid)?

We have adjusted Fig S1D (**new Fig EV2B**) and indicated the deleted nucleotides in Exon 8 that mediate a frameshift of the RNF43 coding region. We also extended the figure legend to explain this in more detail. Indeed, SW480 are aneuploid, however tide analysis of the bulk genotyping of the CRISPR/Cas9-modified cells revealed only two RNF43 variants (V520fs and D516fs).

New Fig EV2B. Sanger sequencing of the PCR amplification products of the mutated RNF43 alleles in SW480 cells. Sequencing results for each mutant allele compared to wild type are shown. The top lines illustrate the wild type RNF43 sequence of nucleotide (nt) 1543-1570. The bottom lines represent the two different RNF43 frameshifts acquired after CRISPR/Cas9 modulation; V520fs (-2 nt) and D516fs (-8 nt).

12. many figure legends are too brief and should be improved. It was difficult to follow what was being done in some of these figures. In general the writing should be improved given that this manuscript presents a convoluted/unexpected mechanism/story.

We adapted the figure legends and Material and Methods section. Furthermore, we extended the discussion of our results based on the referee's suggestions. We thank the reviewer for valuable input with which we were able to amend our manuscript.

References

- Knippschild U, Kruger M, Richter J, Xu P, Garcia-Reyes B, Peifer C, Halekotte J, Bakulev V, Bischof J (2014) The CK1 Family: Contribution to Cellular Stress Response and Its Role in Carcinogenesis. *Frontiers in oncology* 4: 96
- Koo BK, Spit M, Jordens I, Low TY, Stange DE, van de Wetering M, van Es JH, Mohammed S, Heck AJ, Maurice MM *et al* (2012) Tumour suppressor RNF43 is a stem-cell E3 ligase that induces endocytosis of Wnt receptors. *Nature* 488: 665-669
- Marin O, Bustos VH, Cesaro L, Meggio F, Pagano MA, Antonelli M, Allende CC, Pinna LA, Allende JE (2003) A noncanonical sequence phosphorylated by casein kinase 1 in beta-catenin may play a role in casein kinase 1 targeting of important signaling proteins. *Proceedings of the National Academy of Sciences of the United States of America* 100: 10193-10200
- Patel P, Woodgett JR (2017) Glycogen Synthase Kinase 3: A Kinase for All Pathways? *Current topics in developmental biology* 123: 277-302
- Stamos JL, Weis WI (2013) The beta-catenin destruction complex. *Cold Spring Harbor perspectives in biology* 5: a007898

Dear Madelon,

Thank you for submitting your revised manuscript for consideration by The EMBO Journal. My apologies for getting back to you with delay. Your amended study was sent back to two of the referees for re-evaluation. Please note that while referee #2 got delayed and was at this time not able to re-assess your work, we have editorially re-evaluated your response to his-her concerns and found them to be reasonable. We did receive comments from referee #1, which I enclose below. As you will see this referee finds that the concerns raised have been sufficiently addressed and is now broadly in favour of publication, pending minor revision.

Thus, in light of all information at hand, we are pleased to inform you that your manuscript has been accepted in principle for publication in The EMBO Journal, pending some minor issues, which need to be adjusted at re-submission.

Please consider the remaining point of referee #1 regarding the PORCN inhibitor data and related claims by either adding additional experimental data or introducing caveats where appropriate or revising the discussion.

In addition we need you to consider a number of minor points related to formatting and data representation, which are listed below.

Further, I will share additional changes and comments from our production team during the next days to be addressed.

Please contact me at any time if you need any help or have further questions.

As you may have noticed, every paper now includes a 'Synopsis', displayed on the html and freely accessible to all readers. The synopsis includes a 'model' figure as well as 2-5 one-short-sentence

bullet points that summarize the article. I would appreciate if you could provide the bullet points.

Thank you for giving us the chance to consider your manuscript for The EMBO Journal. I look forward to your final revision.

Again, please contact me at any time if you need any help or have further questions.

Kind regards,

Daniel

Daniel Klimmeck PhD
Editor
The EMBO Journal.

>> Please provide up to five keywords for your study.

>> Add Author Contributions for K.B. .

>> Re-check main text callouts for Figures 2B versus 2A (order) and S5.

>> Please add a ToC to the appendix file as a first page.

>> Rename the current 'Declaration of Interests' section to 'Conflict of Interest'

>> Update the Author Checklist with the human consent information.

>> Introduce a separate 'Statistical analysis' section in the Material & Methods part summarizing the strategies applied.

Please see also our instructions to authors

The revision must be submitted online within 90 days; please click on the link below to submit the revision online before 3rd Sep 2020.

Referee #1:

The authors have carefully addressed most of the points and suggestions raised in the original review.

One point is still important to clarify:

In the response to reviewers, the authors state "Of note, prolonged treatment with PORCN inhibitors does not allow survival of onco-RNF43 organoids. The reasons for this are unclear." However in the text the authors state: "Taken together, onco-RNF43 expression confers resistance of human colon organoids to PORCN inhibitor by promoting downstream b-catenin-mediated transcription."

The authors results clearly show a differential sensitivity to PORCN inhibitors between WT organoids and those expressing RNF43-Onc mutants, however if they ultimately are sensitive to the inhibitors, this should be clearly stated and showed. This is especially important given that the authors discuss a possible "contraindication" to PORCN inhibitors or other strategies to inhibit Wnt signalling. If the cells are still reliant (albeit less) on Wnt and Rspo ligands and in the end stop growing in the presence of PORCN inhibitor, will this be a "contraindication"?

Dear Madelon,

This is just another brief note on your manuscript EMBOJ-2019-103932 as we received delayed re-review input from referee #2, stating his-her agreement to our decision to proceed with acceptance of the revised work.

Best regards,
Daniel

Daniel Klimmeck, PhD
Editor
The EMBO Journal

Additional input referee #2:

I am very sorry for my inability to get this done in a timely fashion. I concur with your decision to move ahead with the manuscript.

The authors performed the requested changes.

Dear Madelon,

Thank you for submitting the revised version of your manuscript. I have now evaluated your amended manuscript and concluded that the remaining minor concerns have been sufficiently addressed.

Thus, I am pleased to inform you that your manuscript has been accepted for publication in the EMBO Journal.

Please note that it is EMBO Journal policy for the transcript of the editorial process (containing referee reports and your response letter) to be published as an online supplement to each paper.

Also in case you might NOT want the transparent process file published at all, you will also need to inform us via email immediately. More information is available here:

http://emboj.embopress.org/about#Transparent_Process

Please note that in order to be able to start the production process, our publisher will need and contact you regarding the following forms:

- PAGE CHARGE AUTHORISATION (For Articles and Resources)

[http://onlinelibrary.wiley.com/journal/10.1002/\(ISSN\)1460-2075/homepage/tej_apc.pdf](http://onlinelibrary.wiley.com/journal/10.1002/(ISSN)1460-2075/homepage/tej_apc.pdf)

- LICENCE TO PUBLISH (for non-Open Access)

Your article cannot be published until the publisher has received the appropriate signed license agreement. Once your article has been received by Wiley for production you will receive an email from Wiley's Author Services system, which will ask you to log in and will present them with the appropriate license for completion.

- LICENCE TO PUBLISH for OPEN ACCESS papers

Authors of accepted peer-reviewed original research articles may choose to pay a fee in order for their published article to be made freely accessible to all online immediately upon publication. The EMBO Open fee is fixed at \$5,200 (+ VAT where applicable).

We offer two licenses for Open Access papers, CC-BY and CC-BY-NC-ND.

For more information on these licenses, please visit: <http://creativecommons.org/licenses/by/3.0/> and http://creativecommons.org/licenses/by-nc-nd/3.0/deed.en_US

- PAYMENT FOR OPEN ACCESS papers

You also need to complete our payment system for Open Access articles. Please follow this link and select EMBO Journal from the drop down list and then complete the payment process:

https://authorservices.wiley.com/bauthor/onlineopen_order.asp

On a different note, I would like to alert you that EMBO Press is currently developing a new format for a video-synopsis of work published with us, which essentially is a short, author-generated film explaining the core findings in hand drawings, and, as we believe, can be very useful to increase visibility of the work.

Please see the following link for a representative example:

http://embopress.org/video_EMBOJ-2014-90147

Finally, we have noted that the submitted version of your article is also posted on the preprint platform bioRxiv. We thus appreciate if you could alert bioRxiv on the acceptance of this manuscript at The EMBO Journal in order to allow for an update of the entry status. Thank you in advance!

If you have any questions, please do not hesitate to call or email the Editorial Office.

Kind regards,

Daniel

Daniel Klimmeck, PhD
Editor
The EMBO Journal
EMBO
Postfach 1022-40
Meyerhofstrasse 1
D-69117 Heidelberg
contact@embojournal.org
Submit at: <http://emboj.msubmit.net>

YOU MUST COMPLETE ALL CELLS WITH A PINK BACKGROUND ↓
PLEASE NOTE THAT THIS CHECKLIST WILL BE PUBLISHED ALONGSIDE YOUR PAPER

Corresponding Author Name: Prof. Madelon Maurice

Manuscript Number: EMBO J-2019-103932R